# Ocean cavity regime shift reversed West Antarctic grounding line retreat in the late Holocene

Daniel P. Lowry [1] ✉, Holly K. Han [2,3], Nicholas R. Golledge [4], Natalya Gomez [5], Katelyn M. Johnson [1] & Robert M. McKay [4]

Recent geologic and modeled evidence suggests that the grounding line of the Siple Coast of the West Antarctic Ice Sheet (WAIS) retreated hundreds of kilometers beyond its present position in the middle to late Holocene and readvanced within the past 1.7 ka. This grounding line reversal has been attributed to both changing rates of isostatic rebound and regional climate change. Here, we test these two hypotheses using a proxy-informed ensemble of ice sheet model simulations with varying ocean thermal forcing, global glacioisostatic adjustment (GIA) model simulations, and coupled ice sheet-GIA simulations that consider the interactions between these processes. Our results indicate that a warm to cold ocean cavity regime shift is the most likely cause of this grounding line reversal, but that GIA influences the rate of ice sheet response to oceanic changes. This implies that the grounding line here is sensitive to future changes in sub-ice shelf ocean circulation.

With a sea-level equivalent ice volume of 5.3 m[1], the potential instability of the West Antarctic Ice Sheet (WAIS) is a concern of global significance in a warming world in terms of associated sea-level and climate impacts[2]. Prior to anthropogenic emissions of greenhouse gases, the Holocene epoch (approximately last 11.7 ka) is generally considered to be a period of relative climate stability, with variations in the global average temperature of +/-0.5 °C[3]. Yet emerging geologic evidence suggests that WAIS was smaller-than-present during the middle Holocene, 8.2–4.2 ka BP[4–7]. In the Ross Embayment (Fig. 1), radiocarbon dating of subglacial till from the Whillans and Mercer ice streams (WIS and MIS, respectively) has indicated WAIS retreated hundreds of kilometers upstream of its current position between 7.5 and 5.3 ka BP[8,9]. The timing of WAIS readvance in this sector has been inferred from radiocarbon input and decay modeling to have occurred within the past 1.7 ka for the WIS, and even more recently for the Kamb ice stream (KIS) and Bindschadler ice stream (BIS), at 1 ka BP and 0.8 ka BP, respectively[10]. But the processes contributing to these extensive changes in grounding line position are unclear, and have been attributed to changes in isostatic rebound rates[4,11,12] and regional climatic

changes[10,13]. A climate-driven ice sheet retreat and readvance implies that WAIS is sensitive to Holocene climate variability, whereas isostatic rebound-driven readvance implies that climate played a second-order control on WAIS dynamics along the Siple Coast during this epoch.

Compared to other regions of WAIS, the Ross Embayment has been relatively stable during the observational era, with a positive mass balance[14,15], though ice sheet model projections indicate this sector is vulnerable to future ocean warming[16]. Contributing to the present stability, the Siple Coast grounding line is buttressed by the Ross Ice Shelf, the largest ice shelf by area in Antarctica. The ice shelf overlies a cold ocean cavity, with basal ice shelf melt rates limited by relatively cold High Salinity Shelf Water (HSSW) produced by local polynyas (blue-shaded areas in Fig. 1a)[17]. Despite a number of deeper bathymetric troughs in the outer continental shelf (Fig. 1b), intrusions of relatively warm modified Circumpolar Deep Water (mCDW) are limited in reaching the Siple Coast grounding line under a modern bed configuration[18]. mCDW is responsible for the high basal melt rates observed in the Amundsen Sea sector of WAIS[19], and it is possible that it may have interacted with WAIS ice streams in the past when the bed

[1]Department of Surface Geosciences, GNS Science, Lower Hutt, New Zealand. [2]Fluid Dynamics and Solid Mechanics Group, Los Alamos National Laboratory, Los Alamos, NM, USA. [3]Jet Propulsion Laboratory, California Institute of Technology, Pasadena, CA, USA. [4]Antarctic Research Centre, Victoria University of Wellington, Wellington, New Zealand. [5]Department of Earth and Planetary Sciences, McGill University, Montréal, QC, Canada. ✉e-mail: d.lowry@gns.cri.nz

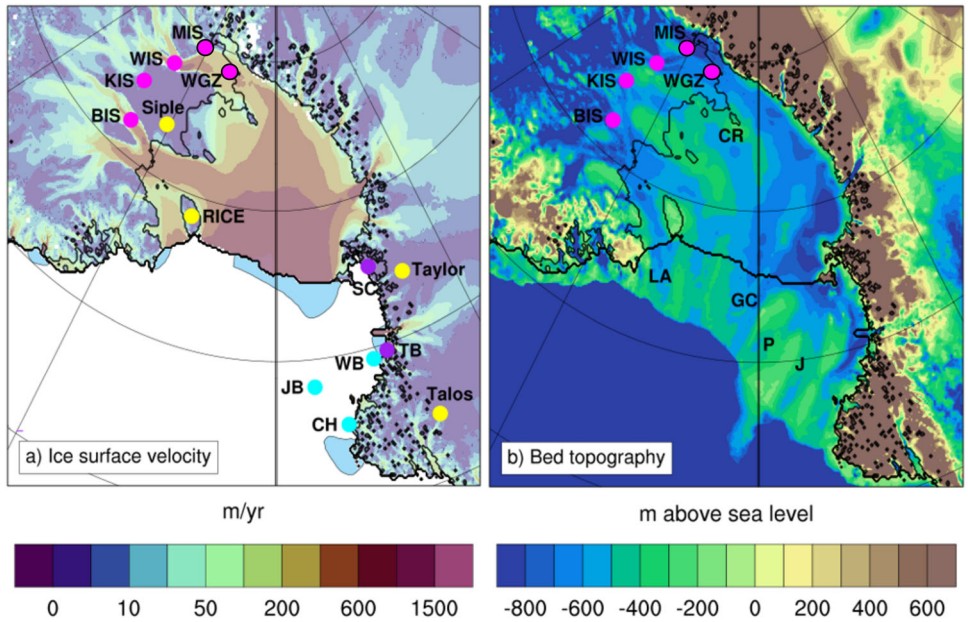

**Fig. 1 | Modern surface ice velocity and bed topography of the Ross Sea region of Antarctica. a** Modern ice surface velocity (m/yr) and **b** bed topography (m above sea level) of the Ross Sea region of Antarctica. Colored circles indicate locations of proxy records, including subglacial sediments (magenta), ice cores (yellow), relative sea level reconstructions (purple), and marine sediments (cyan). Light blue shading indicates locations of polynyas in the Ross Sea. Magenta circles outlined in black indicate the age constraints are from direct measurements, whereas unoutlined magenta circles indicate the age constraints are modeled. WIS Whillans Ice Stream site, KIS Kamb Ice Stream site, BIS Bindschadler Ice Stream site, WGZ Whillans Grounding Zone site, Siple Siple Dome ice core, RICE Roosevelt Island Climate Evolution ice core, Taylor Taylor Dome ice core, Talos Talos Dome ice core, SC Scott Coast, TB Terra Nova Bay, WB Wood Bay marine core, JB Joides Basin marine core, CH Cape Hallett marine core, LA Little America Basin, GC Glomar Challenger Basin, P Pennell Basin, J Joides Basin.

topography was lower[20], or because of reduced polynya activity[21]. Previous studies have invoked the transition from a warm to cold ocean cavity to explain a slowing down of grounding line retreat during the deglaciation[22], but could such a regime shift potentially explain the reversal in grounding line migration that occurred from the mid-to-late Holocene?

An alternative explanation for WAIS retreat and advance was first suggested in ref. 4 for the Weddell Embayment, which noted that low modern bed uplift rates could be explained by extensive grounding line retreat during the last deglaciation when the bed was lower, with subsequent isostatic rebound causing regrounding of the ice sheet. Kingslake et al.[11] further explored this mechanism for the wider WAIS, including the Ross Embayment, using process-based ice sheet model simulations. Their ice sheet model adopts a simple 1D Earth deformation model with an elastic plate overlain by a viscous mantle, and the results are particularly sensitive to the value used for mantle viscosity. Only simulations with mantle viscosity values within the range of 5e20–1e21 Pa s produce a rate of isostatic rebound that allows this isostatic rebound-driven regrounding of WAIS to occur in the Holocene; below (above) this range, the bed topography adjusts too rapidly (slowly) in response to ice sheet deglaciation. Although estimations of the average West Antarctic upper mantle viscosity tend to be lower than 5e20 Pa s[23,24], there is high spatial heterogeneity in Earth structure across the tectonic divide of the Ross Embayment with orders of magnitude of uncertainty[25]. Because these simulations were performed prior to more definitive age estimations of marine exposure to the subglacial till[8–10], there was no age target to constrain the model, and the model predicted retreat and advance occur thousands of years earlier than the recently reconstructed ages. Does delayed rebound still explain the paleoclimate data within this more narrowly constrained time range?

The goal of this study is to test these competing hypotheses and better understand the interaction between the ice sheet, the solid Earth, and the ocean as the Ross Ice Shelf developed. To do so, we use an ensemble of ice sheet model (ISM) simulations with both a simplified viscoelastic deformation model and a global glacioisostatic adjustment (GIA) model that captures ice-ocean gravity, Earth rotational effects and viscoelastic Earth deformation (GRD)[26,27]. Our ISM simulations are performed using the Parallel Ice Sheet Model[28,29], following the same approach as in ref. 30, with some modifications (see "Methods"). We then apply the ice thickness outputs from the ISM as input to a 1D GIA model incorporating a range of radially varying Earth structures to represent both GIA and local sea-level change. Finally, we apply an iterative coupling between the ice sheet and GIA models[31] to consider the feedback of GIA on the late Holocene ice-sheet readvance. More detailed descriptions of both models are provided in "Methods". Our simulations are focused on assessing the relative contributions of GIA and ocean thermal forcing to Holocene grounding line migration in the Ross Embayment.

## Results

### Ice sheet response to ocean forcing with simplified viscoelastic Earth deformation

Our ISM simulations are run from 40 to 0 ka BP. Details of the environmental forcings used in these experiments and ice sheet model validation are provided in the Supplementary Information (Supplementary Figs. S1–S3). Bed topography evolves based on the ice sheet load history and the solid Earth parameters used in the simple viscoelastic Earth deformation model, including mantle viscosity, mantle density, and lithosphere flexural rigidity, which influence GIA rates (Supplementary Figs. S4 and S5)[32,33]. This is the same viscoelastic deformation scheme employed in ref. 11, but with a corrected numerical implementation for the elastic component[34]. The surface climate forcing is in the form of temperature and precipitation anomalies relative to 0 ka BP derived from the WAIS Divide ice core[35], applied as a percentage anomaly to the present-day climatology[36]. The standard ocean forcing is derived from temperature and salinity anomalies of the TraCE-21ka climate model simulation[37,38], averaged at

500 m depth. These surface climate and ocean fields are updated every 100 years.

While the surface climate evolution of Antarctica is relatively well known over the last deglaciation from the distribution of Antarctic ice core records, ocean circulation under ice shelves is poorly constrained. Different approaches exist for the implementation of ocean forcing in paleo-ice sheet simulations, including deriving ocean temperatures from ice core records using response time functions[34], benthic ocean temperature reconstructions[39], and ice sheet-proximal ocean temperature and salinity from global climate models[30,40]. Global climate models allow for spatial variability in ocean forcing for different regions of the ice sheet, which is advantageous. However, the TraCE-21ka simulation, which we use here, has been suggested to show bias with respect to the oceanic response to prescribed deglacial meltwater[41,42]. More specifically, because TraCE-21ka does not resolve the ice shelf cavity and relevant processes during its formation, we also performed modified ocean forcing experiments with ocean temperature anomalies imposed in the Ross Sea region during the Holocene, starting from 7 ka and finishing at the time of proposed ice sheet readvance (1.6 ka BP) in ref. 10. The start date is based on the findings of Ashley et al.[22], who note a peak in biomarker isotopes indicating enhanced meltwater derived from the Ross Sea between 8 and 4.5 ka BP, as well as cosmogenic nuclide studies of glacial erratics in the Transantarctic Mountains indicating the majority of deglaciation in the Ross Sea occurred between 9 and 3.5 ka BP[43–46]. As proxies from the Ross Sea region support enhanced intrusion of mCDW from 6 to 2.8 ka BP and 1.6 to 0.7 ka BP[47], the aim of these sensitivity experiments is to assess the grounding line response to plausible changes in Holocene oceanic forcing.

Predicted grounding line migration in the Ross Embayment shows strong sensitivity to both mantle viscosity and ocean thermal forcing (Fig. 2). Higher mantle viscosity (e.g., 1e21 Pa s) is associated with earlier and more extensive ice sheet retreat for a given ocean forcing. In comparison to mantle viscosity, lithosphere flexural rigidity exhibits less influence on grounding line migration, whereas increasing mantle density delays grounding line retreat and limits readvance (Supplementary Figs. S4 and S5). Simulations with low mid-Holocene ocean temperature anomalies (i.e., +0.3 °C) show instances of regrounding in the Western Ross Sea in regions of raised bed topography due to an expanded ice shelf and uplifting bed. Simulations with higher ocean temperature anomalies (i.e., +0.5) show a later minimum grounding line extent, regardless of mantle viscosity. At the lower end of the mantle viscosity range (e.g., 5e20–7.5e20 Pa s), these higher ocean temperature anomalies are required to retreat beyond some of the WAIS ice stream proxy locations. However, in nearly all instances, the retreat of the KIS is more limited than the retreat of WIS, MIS, and BIS.

Radiocarbon dating based on ramped pyrolysis of low-carbon grounding line-proximal and subglacial sediments[8,9] and input and decay modeling[10] offer age constraints for marine exposure of the ice stream locations which are used here to assess the model simulations (Fig. 3). At Whillans Grounding Zone (WGZ), close to the modern grounding line, Venturelli et al.[8] suggest a mid-Holocene grounding line retreat occurring at 7.2 ka BP, with an uncertainty range from 7.5 to 4.8 ka BP. Simulations with mantle viscosity of 5e20 and 7.5e20 Pa s show a consistent timing of retreat at this site with an ocean temperature anomaly of at least +0.4 °C. Even without any imposed Holocene ocean warming, simulations using a mantle viscosity of 1e21 Pa s show retreat that is >1.5 ka too early. In all, 250 km further upstream at the MIS site (see Fig. 1), Venturelli et al.[9] constrain retreat to 6.3 ka BP ± 1 ka, indicating a less rapid retreat than the simulation shown in Fig. 2i (MV of 1e21 Pa s; +0.5 °C), but more rapid than that in Fig. 2h (MV of 7.5e20 Pa s; +0.5 °C). More rapid grounding line retreat of WIS and MIS is possible with mantle viscosity lower than 1e21 Pa s with either earlier onset of anomalous ocean warming, or with higher

temperatures (Supplementary Fig. S6); though in these simulations, ice shelf collapse occurs if ocean temperatures exceed +0.5 °C for >1000 ka, hence we consider +0.5 °C to be the upper limit of plausibility for the long-term average over multi-millennial timescales.

Using a two-phase model of radiocarbon input and decay, Neuhaus et al.[10] estimate both the timing of the retreat and advance phase for upstream sites of WIS, KIS, and BIS. Only our simulations with ocean temperature anomalies of at least +0.5 °C show sufficient retreat at these sites. As the ocean forcing anomaly is removed at 1.6 ka, the overall ice sheet advance is greater with increasing ocean temperature anomalies, and the rate of advance decreases with increasing mantle viscosity because the topographic high points of the Siple Coast are relatively lower (Supplementary Fig. S7). Notably, the radiocarbon modeling indicates differences in behavior between the various ice streams, with later retreat and advance of KIS and BIS relative to WIS. If we apply the anomalous forcing to 1 ka BP, the timing of our modeled readvance of BIS and KIS is more consistent with the inferred ages of Neuhaus et al.[10] (Supplementary Fig. S6). Overall, these experiments with simple treatment of Earth deformation suggest that although the delayed rebound mechanism of retreat is inconsistent with the proxy data, Earth structure does influence how sensitively the ice sheet responds to changes in ocean forcing, both in terms of grounding line retreat and advance.

## Gravitational, rotational, and deformational (GRD) effects in the Ross Sea region

At ice sheet margins, a number of factors influence the relative sea level, which in turn impacts grounding line dynamics[48,49]. While the ISM simulations described above account for the viscoelastic deformation of the solid Earth in response to changes in ice sheet load distribution, the treatment of deformation is simplified to an elastic plate overlain by a viscous half space[32,33]. As an ice sheet loses mass, loss of ice-ocean gravity and uplift of the solid Earth lead to a local sea-level fall that can slow grounding line retreat or trigger grounding line advance, while ice mass gain leads to a local sea-level rise that can reduce grounding line advance or trigger grounding line retreat[48]. Furthermore there is a feedback of global ice load-induced perturbations in the orientation of the Earth's rotation vector that has contributed to variable relative sea level changes around Antarctica since the Last Glacial Maximum[50,51]. Thus, it is important to consider the above-mentioned GRD effects in response to ice-sheet changes in a self-consistent manner in assessing the late Holocene ice-sheet advance.

To properly account for GRD effects on sea level, we ran global GIA model simulations from 40 to 0 ka BP using an initial bed topography and ice sheet thickness output from the ISM, updated every 100 years (see "Methods"). This improves upon the standalone ISM simulations in terms of the computation of deformation as the GIA model is based on normal-mode theory in which each mode of deformation wavelength has a distinct relaxation time[48], whereas the ISM computes deformation in less sophisticated manner[33]. We consider two end-member cases of Earth structure relevant for the Ross Sea region. The first GIA model adopts a relatively weak Earth structure (hereafter referred to as GIAWE) and is based on the findings of ref. 23. The lithosphere thickness is 60 km, the upper mantle viscosity ranges from 1e18 to 1e19 Pa s, and the lower mantle viscosity is 1e22 Pa s. The second GIA model adopts a relatively strong Earth structure (hereafter referred to as GIASE) that is more consistent with the ISM simulations with simple Earth deformation described above, and is within the range of estimations for the Siple Coast[24,25]. The lithosphere thickness is 90 km, the upper mantle viscosity is 5e20 Pa s, and the lower mantle viscosity of 5e21 Pa s.

Figure 4 shows predictions of bed topography changes from the ISM with simple viscoelastic deformation using a mantle viscosity value of 7.5e20 Pa s, and GIA model simulations adopting GIAWE and GIASE Earth structures. All predictions adopt the same ice cover

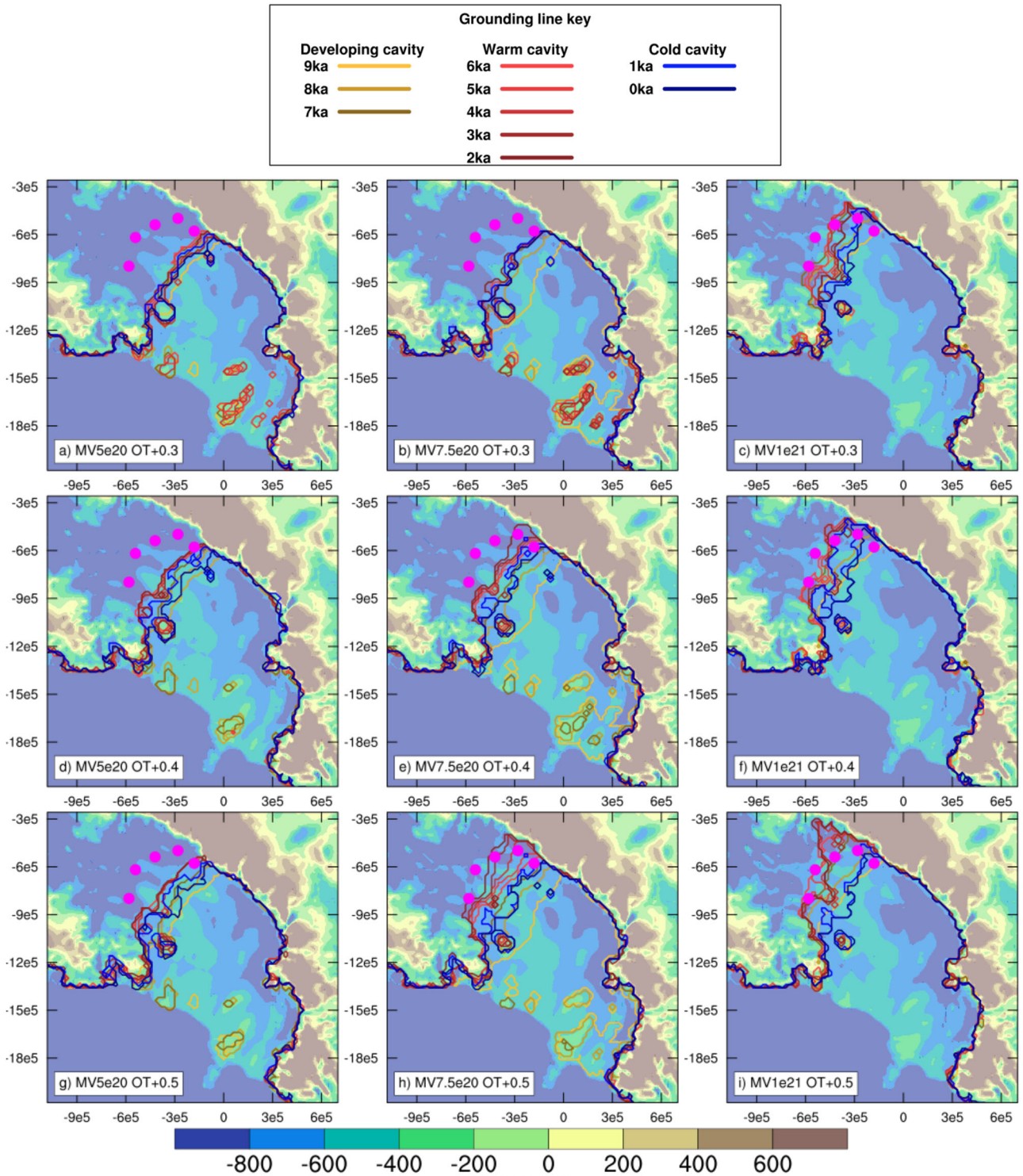

**Fig. 2 | Predicted Holocene grounding line migration with varying ocean forcing and mantle viscosity. a–i** Predicted bed topography (m above sea level) at 0 ka BP in the Ross Sea sector from the ice sheet model experiments using simple viscoelastic Earth deformation with varying mantle viscosity (MV; units of Pa s) and mid-Holocene ocean temperature anomalies (OT; units of °C). The grounding line position is shown every 1000 years from 9 to 0 ka BP, darkening in scale, with color changes indicating transitions in the ocean forcing: standard TraCE-21ka forcing from 9 to 7 ka BP (gold), an anomalous ocean warming phase from 7 to 2 ka BP (red), and a return to the standard TraCE-21ka forcing (blue). Magenta circles indicate the West Antarctic subglacial sediment sites.

changes. We focus on this particular simulation because it is the most consistent with the timing of WIS and MIS retreat reconstructed in refs. 8 and 9, as well as the timing of WIS, KIS, and BIS retreat and advance modeled in ref. 10, while maintaining a stable ice shelf. A main difference between the ISM and GIA models during the retreat phase

(7–2 ka BP) is that the bed uplift is more localized to the region of grounding line retreat, where the greatest decrease in grounded ice volume has occurred (Supplementary Fig. S8), in the GIA models. Uplift is higher upstream of the modern grounding line in the GIAWE relative to the ISM (Supplementary Fig. S9), but strong subsidence

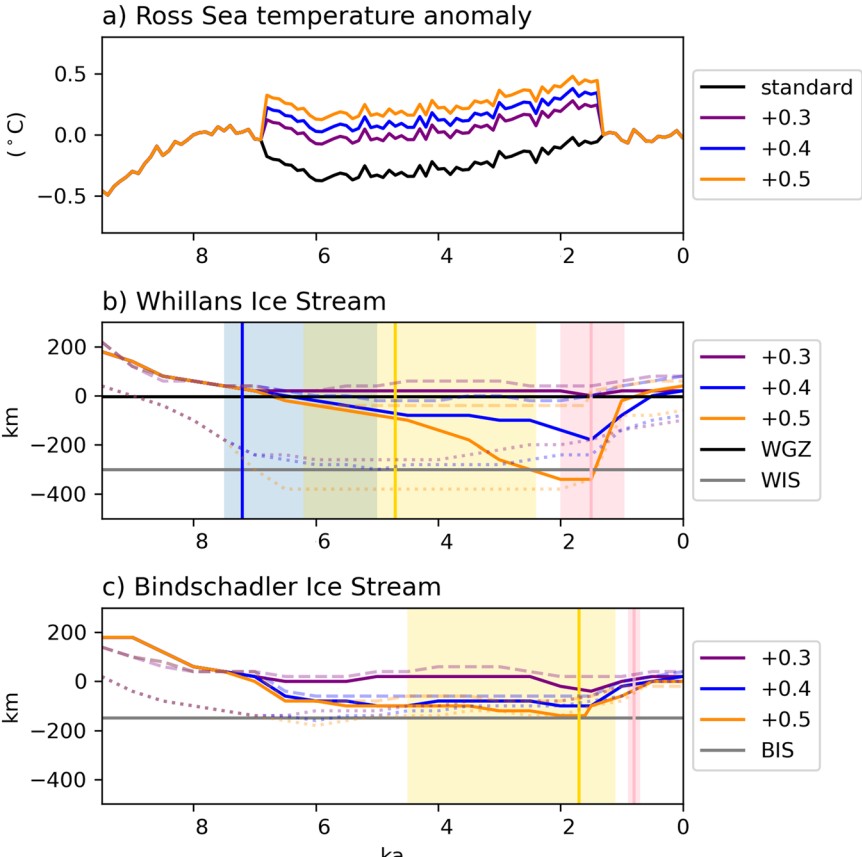

**Fig. 3 | Comparison of modeled grounding line position of West Antarctic Ice Sheet ice streams to proxy reconstructions. a** Standard TraCE-21ka and modified ocean forcing for the Ross Sea sector. Distance of ice sheet grounding from the modern grounding line for the (**b**) Whillans and **c** Bindschadler ice streams (WIS and BIS, respectively) for simulations with a mantle viscosity of 7.5e20 Pa s. These ice stream transects are defined based on the central stream line of the present-day ice sheet, with the grounding line defined as the first instance of a grounded ice sheet along this transect. Horizontal lines correspond to the proxy sites. Dotted (dashed) lines show simulations with higher (lower) mantle viscosity, 1e21 Pa s (5e20 Pa s). The vertical blue line shows the radiocarbon age for retreat at Whillans Grounding Zone (WGZ) from ref. 8, with blue shading indicating age uncertainty. The vertical yellow and pink lines respectively indicate the modeled age for retreat and advance at the WIS and BIS sites from ref. 10 with colored shading indicating age uncertainty.

occurs under the ice shelf due to the collapse of the peripheral bulge. The GIASE, in contrast, shows less uplift in general than in the ISM during the retreat phase (Supplementary Fig. S10).

Because of the lack of bedrock exposure on the Siple Coast, it is difficult to determine past rates of sea-level change or modern uplift rates to assess these models. The best proxy reconstructions of relative sea level available are from the Transantarctic Mountains on the western Ross Sea side of the embayment[52–54], across a tectonic divide[20], and likely with much lower mantle viscosity that is similar to the Amundsen Sea Embayment[24,25]. On the Scott Coast, a reconstruction from marine shells, seal skin, and elephant seal remains indicates a relatively gradual decrease in sea level from 6.5 ka[52]. The GIAWE is most consistent with this reconstruction. At Terra Nova Bay, radiocarbon dates of penguin guano and remains, shells, and seal skin from raised beaches indicate a more rapid decrease in relative sea level at this time[53]. At both sites, the ISM overestimates the relative sea level in the mid-Holocene, highlighting the marked improvement gained from using a global GIA model that accounts for GRD effects as compared to a simple viscoelastic deformation model.

During the advanced phase (2–0 ka BP), the ISM and GIASE show modest bed uplift of <30 m, but with areas of localized subsidence upstream of the modern Siple Coast grounding line of WIS and BIS due to the increased ice loading from regrounding of the ice sheet. The subsidence in GIAWE is of higher magnitude and more extensive than in the GIASE and ISM. Also notable is the bed uplift from a peripheral bulge under the modern ice shelf in GIAWE, which is of a magnitude to

potentially impact sub-ice shelf ocean circulation. Lastly, it is important to note that none of the models predict delayed bed uplift during this advance phase, as would be expected if this was the key driver of ice sheet advance.

### Relative contributions of ocean forcing and GRD effects to ice sheet advance

Ocean forcing and glacioisostasy both influence grounding line migration, and it is difficult to isolate their relative contributions. The impact of the ocean forcing changes on the ice sheet is shown as a sequence of ice stream transects in Fig. 5 for the MV7.5e20 Pa s simulation with an ocean temperature anomaly of +0.5 °C. Basal melt rates and horizontal ice flux increase as the ice sheet thins, retreats, and ungrounds. The basal melt rates that cause the grounding line retreat are less than 50% of those currently observed in the Amundsen Sea sector, where bulk melt rates reach up to 20 m yr⁻¹ (see ref. 55). The relative cooling that occurs when the anomalous ocean forcing is removed reduces basal melt rates of both ice streams while the ice flux is still high. This leads to ice shelf thickening, regrounding, and grounding line advance.

It is notable that the basal ice shelf melt rate changes are an order of magnitude greater than variations in surface mass balance reconstructed at WAIS Divide (i.e., <0.2 m yr⁻¹) (Supplementary Fig. S1). The WAIS Divide record shows a decreasing snow accumulation trend coinciding with the time of WAIS advance[35]. Shallow firn-cores at Siple Dome show lower snow accumulation relative to other parts of WAIS,

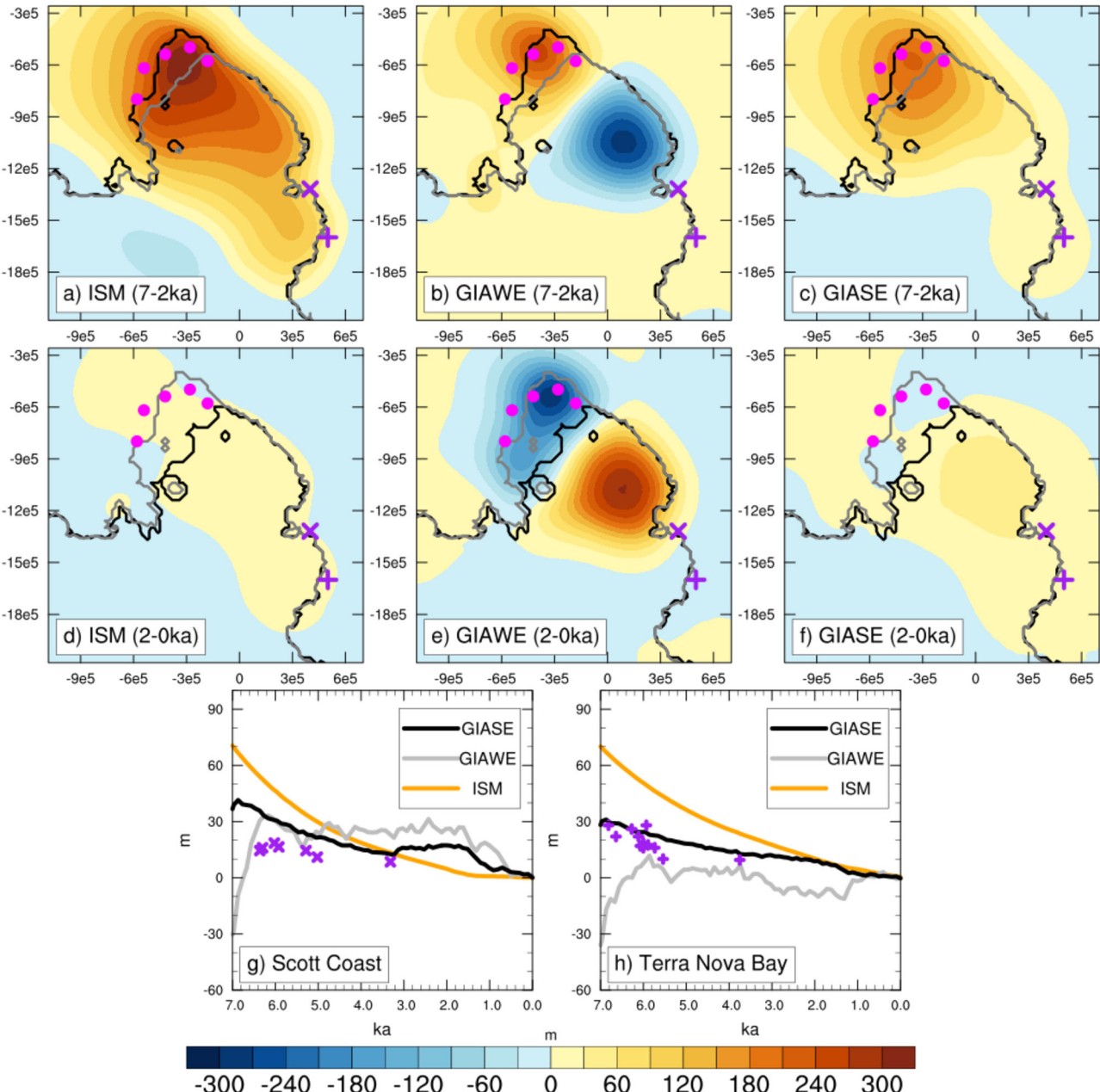

**Fig. 4 | Bed topography changes and relative sea level predicted by the ice sheet and glacioisostatic adjustment (GIA) models. a–c** Bed topographic change (m) for the ice sheet model (ISM) and GIA models with weak Earth structure (GIAWE) and strong Earth structure (GIASE) during the retreat phase of the MV7.5e20 + 0.5 °C simulation (solid orange line in Fig. 3). All simulations adopt the same ice history, shown in Fig. 2h (grounding line migration) and Supplementary Fig. S8 (ice thickness change). **d–f** Bed topographic change for the same model simulation, but during the advance phase. The purple markers indicate the proxy site locations. **g, h** Relative sea level from the three models and reconstruction. For the ISM, relative sea level is determined relative to the global mean sea-level forcing (Supplementary Fig. S1), whereas the GIA models account for the gravitational, rotational, and deformational (GRD) effects of the Northern Hemisphere Ice Sheet deglaciation using the ICE-5G dataset[89].

with annual average rates of 0.155 m yr⁻¹ since 1955[56]. Furthermore, internal reflective horizons detected by radio-echo sounding have been used to infer that surface mass balance was 18% higher relative to modern over the WAIS Divide at 4.7 ka BP[57], coinciding with the radiocarbon-modeled retreat of WIS, KIS, and BIS[10]. The most efficient addition of mass into the ice sheet-ice shelf system to drive the readvance is therefore a reduction of basal melt rates and marine refreezing, which in our simulation leads to an instantaneous change of ice shelf mass balance of multiple meters per year from the retreat phase to the advance phase. Surface mass balance would require a greater than 30 × increase from present to be comparable to the basal mass

balance change from the reduction of the peak melt rates at BIS simulated here.

In the ISM, as the ice sheet is advancing, the bed below the ice shelf is uplifting in response to the decreased ice load from the previous ice sheet retreat (Fig. 5). Though, in comparison, the GIAWE and GIASE simulations show localized subsidence upstream of the modern grounding line position (Fig. 4e, f). To assess the extent to which the bed uplift in the ISM is contributing to the readvance, we run additional ice sheet advance experiments with the ISM using different bed topographies (Fig. 6). These experiments are initiated from a near-minimum grounding line extent (2.0 ka BP) and are

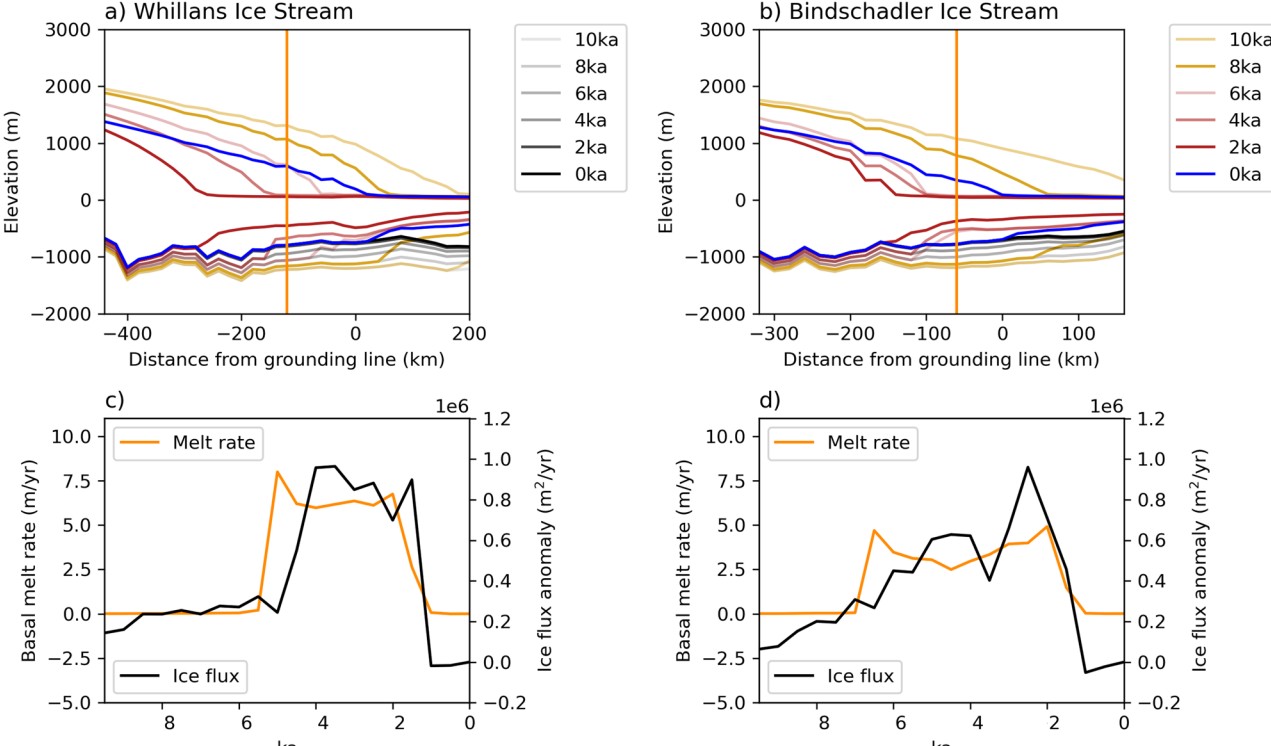

**Fig. 5 | Ice stream and ice shelf responses to changes in ocean forcing.**
**a**, **b** Vertical transects of the Whillans and Bindschadler ice streams every 2 ka from 10 to 0 ka BP from the ice sheet model in Fig. 4. The lines darken in color through time, with color changes corresponding to changes in the ocean forcing: standard TraCE-21ka forcing from 9 to 7 ka BP (gold), an anomalous ocean warming phase

from 7 to 2 ka BP (red), and a return to the standard TraCE-21ka forcing (blue). Bed topography through time is represented by the gray lines, also darkening through time. **c**, **d** Basal melt rate (solid line) and ice flux anomaly relative to 0 ka (dotted line) at the transect location indicated by the vertical orange line in (**a**, **b**).

run forward through the ocean temperature anomaly reduction. In the first case, the bed remains fixed in its current position (i.e., no bed uplift occurs). In the other cases, we apply the bed topography changes from the GIAWE and GIASE simulations, applied as anomalies to the 2.0 ka bed topography, using the iterative coupling procedure of ref. 31.

The ice sheet response to ocean cooling in the advance experiments is similar in all of the experiments in terms of both extent and timing of grounding line advance (Fig. 6). For the WIS, MIS, and KIS, the ISM with simple viscoelastic Earth deformation, GIAWE, and GIASE show grounding line advance to near-modern position within 400 years, with advance of BIS delayed by another 200 years. This demonstrates the lower sensitivity of the BIS to changes in ocean forcing than the WIS, but can only partly account for the difference in timing indicated by the radiocarbon input and decay model (Fig. 3c). We discuss other possibilities for differences in ice stream behavior in the following section. Only the fixed bed simulation deviates from the other experiments for WIS, with a slower rate of grounding line advance. The high uplift in the GIAWE simulation grounds portions of the Ross Ice Shelf, highlighting the particular sensitivity to this weak Earth structure and suggesting such Earth structure is unrealistic for the Siple Coast of WAIS as well as the importance of considering the laterally varying (3D) Earth structure in Antarctica for robust comparison between model results and observations. Overall, these simulations demonstrate that the ice sheet response to relative cooling is robust and consistent with the timescales of change inferred from the age constraints of the radiocarbon input and decay model, regardless of the bed topography changes over the past 2.0 ka. This result supports the Neuhaus et al.[10] hypothesis that modest oceanic cooling is the driver of late Holocene ice sheet readvance in this sector of the WAIS.

## Discussion

Of the two hypotheses we test, our results indicate that a warm-to-cold ocean cavity regime shift is a more plausible mechanism for the reversal of grounding line retreat, with the acknowledgment that GIA is an important influence on how sensitively the ice sheet responds to changes in ocean forcing. The ISM only simulates grounding line retreat to the WIS, MIS, KIS, and BIS sites under high mid-Holocene ocean temperature anomalies. When these higher ocean temperature anomalies cease, extensive and rapid grounding line readvance occurs, on the timescale of hundreds of years. In contrast, under low and no mid-Holocene ocean warming conditions, grounding line retreat to the WIS, MIS, KIS, and BIS sites can only occur with relatively high mantle viscosity, inconsistent with estimations of the Siple Coast region[24,25]. In these simulations, the retreat occurs earlier than the radiocarbon ages of refs. 8 and 9, and the readvance is more gradual than inferred from the radiocarbon model of ref. 10, occurring over thousands of years. With simple viscoelastic Earth deformation, lower mantle viscosity is associated with a more rapid ice sheet response time to oceanic cooling because the late Holocene bed topography is relatively higher (Supplementary Fig. S7). This implies that the isostatic rebound that occurs prior to the late Holocene oceanic cooling is a second-order control on WAIS readvance. Accounting for GRD effects with WAIS-like Earth structure, local bed subsidence occurs as the ice sheet regrounds, meaning that prolonged isostatic rebound is unlikely to be the main driver of ice sheet advance.

Regional paleoclimate reconstructions support a warm ocean cavity during the ice sheet retreat phase (Fig. 7). Methanesulfonic acid (MSA) concentrations from ice cores indicate past changes in biological activity, sea ice extent, and polynya activity[58]. At Taylor Dome, MSA is low in the early Holocene, consistent with widespread sea ice cover and less open ocean area. In the middle Holocene, from 7.1 ka BP,

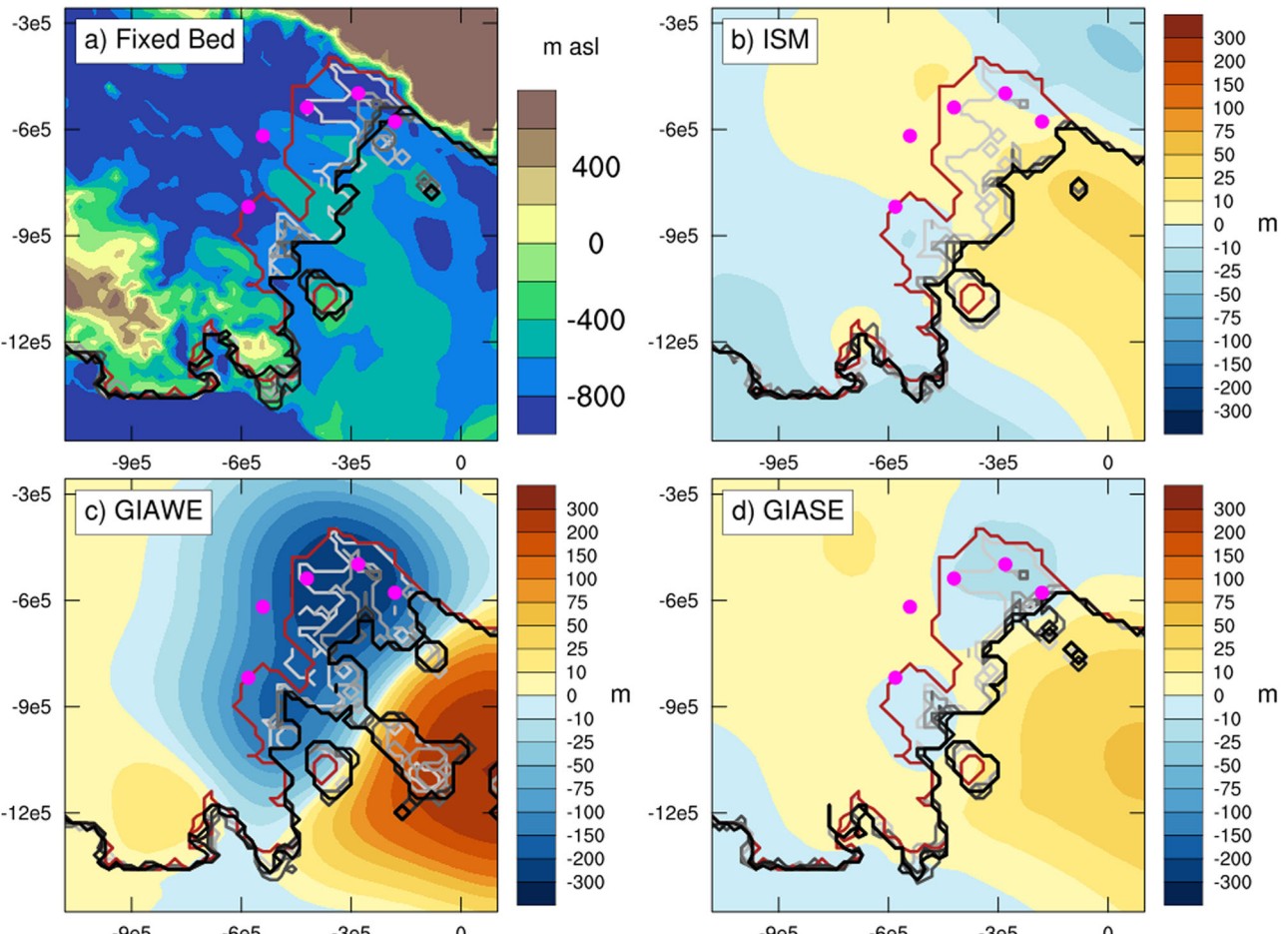

**Fig. 6 | Influence of bed topographic changes on ice sheet response to oceanic cooling.** Predicted grounding line migration of the ice sheet advance experiments using different bed topography changes. The minimum grounding line extent at 1.6 ka BP is shown by the dark red line, and position changes every 200 years are shown, darkening in scale from gray to black. **a** Fixed bed topography at 2 ka BP shown in m above sea level; **b** bed topography change (m) from 2 to 0 ka BP of the ice sheet model (ISM) with simple viscoelastic deformation; **c** bed topography change (m) from 2 to 0 ka BP of the glacioisostatic adjustment model with weak Earth structure (GIAWE); **d** bed topography change (m) from 2 to 0 ka BP of the glacioisostatic adjustment model with strong Earth structure (GIASE).

land-fast sea ice declines in the Western Ross Sea, with peak ocean temperatures at 5.2 ka BP[59], coinciding with retreat of WIS and MIS[8,9]. West Antarctic summer temperatures increase above pre-industrial to a peak at 4.1 ka[60] as WIS continued to retreat, and BIS and KIS retreated[10]. MSA remains low during this warm climate period, which suggests weak polynya activity and reduced overturning. Cavity-resolving regional ocean model simulations have demonstrated that changes in offshore winds, salinity, and overturning influence the delivery of mCDW in modern cold ocean cavities[61–64]. Our ISM simulations place an upper limit of ocean temperature anomalies in the cavity of 0.5 °C averaged over 100 year time periods, though ocean forcing with higher variability may extend this temperature threshold higher. This means that even temporary intrusions of mCDW that increase the average temperature are sufficient for the Holocene grounding line retreat beyond the present day. Such intrusions are more likely to occur under more stratified conditions with a shorter duration of winter sea ice and weaker katabatic winds (Fig. 7c).

A transition to a cold cavity in the late Holocene is also supported by the paleoclimate evidence of the Neoglacial transition from 4.5 ka BP. At the end of the ice sheet retreat phase, negative excursions in deuterium isotopes from fatty acid biomarkers indicate an increase in meltwater flux as the Ross Ice Cavity expanded between 6 and 3 ka[22]. This was interpreted as enhanced production of supercooled ice shelf water feeding surface waters and an increase in sea ice production, indicated by the Taylor Dome MSA (Fig. 7a), but also downstream at

offshore Adelie land[22]. Diatom reconstructions of the Western Ross Sea suggest an increase in katabatic winds, which causes a strengthening of the Ross and Terra Nova Bay polynyas over the last 3.6 ka[21] as well as a strengthening of bottom currents[65]. From regional cooling and stronger katabatic winds, increased production of HSSW from coastal polynyas during the extended sea ice season may have prevented mCDW intrusions (Fig. 7d). This regional climatic change in the Late Holocene essentially represents the transition to the modern oceanic conditions that influence the regional ice sheet behavior today, in which local polynyas flood the ice shelf cavity with cool shelf water[17].

For the Siple Coast grounding line, the ice sheet simulations shown here demonstrate that a regime shift from a warm to cold ocean cavity is capable of driving hundreds of kilometers of advance, consistent with geological evidence over a range of timescales. For example, subglacial precipitates also suggest that millennial-scale oscillations in Southern Ocean temperatures drive ice sheet velocity fluctuations through the addition of subglacial meltwater from the ice sheet interior during warm periods and subglacial freezing during cold periods[66]. Furthermore, centennial-to-millennial scale shifts in the El Niño Southern Oscillation and Southern Annular Mode are observed to have dramatically influenced circum-Antarctic winds and sea ice heterogeneously the Late Holocene, as inferred from marine sediment cores[67,68]. Such shifts would influence heat flux in the cavity, and therefore raises the possibility of grounding line oscillations at these timescales within the Holocene. While the Neoglaciation transition

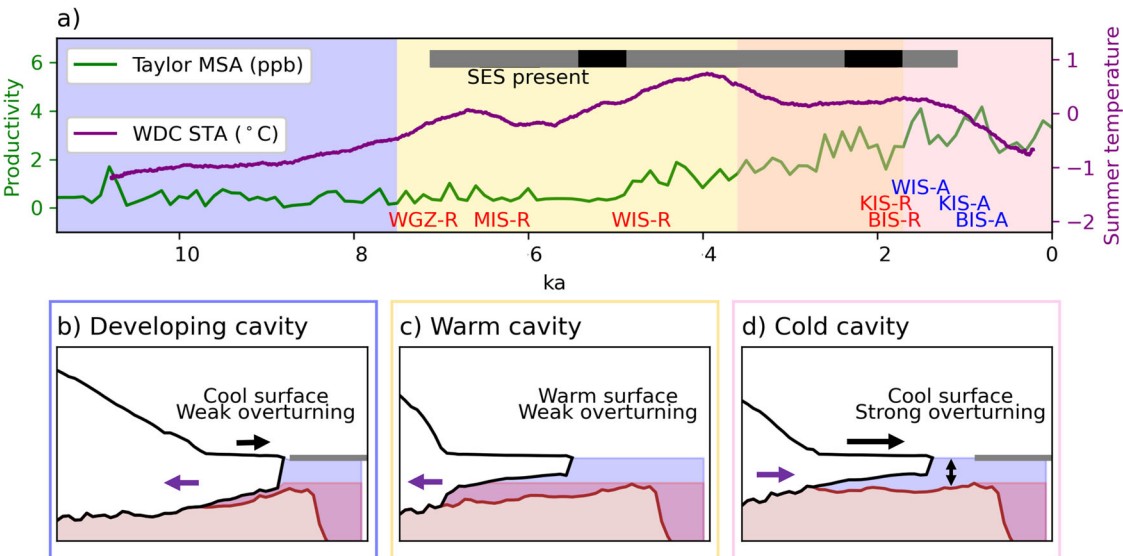

**Fig. 7 | Ocean cavity regime shift as a mechanism for West Antarctic Ice Sheet (WAIS) retreat and readvance. a** Paleoclimate reconstructions from Antarctic ice core records. The gray bar indicates the presence of Southern Elephant Seals (SES) in the Western Ross Sea region of Antarctica, which indicate reduced land-fast sea ice[59]. The black sections denote especially warm periods. The green line shows Methanesulfonic acid concentrations (MSA) at Taylor Dome[58,90], a proxy for marine biological productivity related to polynya activity. The purple line shows a reconstruction of summer temperature anomalies (STA) from the WAIS Divide ice core (WDC) from ref. [60]. The approximate timing of grounding line retreat (-R) and advance (-A) is indicated for the WAIS subglacial sediment sites. The background colors correspond to approximate timings of ice sheet phases: early Holocene retreat to the modern grounding line position (blue), middle Holocene retreat to the minimum grounding line position (yellow), and late Holocene advance to the modern grounding line position (pink). The overlapping yellow and pink represent the uncertainty in the precise timing of cold cavity initiation and with the acknowledgment that regional differences may exist. The timing is based on the long-term increase in Ross Sea polynya efficiency from 3.6 ka BP[21], though centennial-scale oscillations in polynya efficiency and mCDW intrusion do occur through this interval[47,59], until the time of ice stream advance indicated in pink from 1.6 ka BP. **b**–**d** Ice shelf/ice sheet transects along Bindschadler Ice Stream (BIS) for the ice sheet phases above. The gray bar indicates sea ice extent. Red shading indicates the location of mCDW. The purple arrow indicates direction of grounding line change. The size of the black arrow corresponds to the strength of katabatic winds.

represents a state change in Southern Ocean conditions on millennial timescales, evidence of centennial-scale Late Holocene variability in polynya efficiency is also evident[21], with low polynya efficiency during a particularly warm phase from 2.3 to 1.8 ka BP indicated by the presence of elephant seals in the western Ross Sea[59], also coinciding with retreat of BIS and KIS[10]. WIS advances following this Late Holocene warm event, but BIS and KIS advance within the last 0.8 ka BP following strong regional cooling that led to collapse of the seal population (Fig. 7a). The timing of readvance of MIS is unconstrained, though the ice sheet model shows a synchronous timing of advance to WIS.

While our simulations imply a climatic control on Holocene grounding line dynamics, solid Earth rheology does exert influence that is important to consider in interpreting the results presented here. Much attention has been given to modern rapid isostatic rebound occurring in the Amundsen Sea Embayment, with estimated mantle viscosity substantially lower than the global average[23,49]. The Ross Embayment presents a modeling challenge because it spans a tectonic divide, with orders of magnitude difference in both mantle viscosity and lithosphere flexural rigidity across the Transantarctic Mountains and across the western to eastern Ross Sea[24,69,70]. While no single set of parameter values in these ice sheet and 1D GIA models can fully capture these ranges, the ensembles are useful for highlighting that mantle viscosity affects bed topography, which in turn influences ice sheet sensitivity to changes in oceanic conditions (Figs. 2 and 3). In particular, the GIASE simulation shows reasonable migration of the Siple Coast grounding line, whereas the GIAWE does not. This is relevant for the Siple Coast because it has relatively higher mantle viscosity and lithosphere flexural rigidity by West Antarctic standards, meaning that the grounding line here is more sensitive to increases in ocean thermal forcing than other areas of weaker mantle.

Another influence of bed topography is on the sub-ice shelf ocean circulation, with the possibility that the uplift of the bed

contributed to a transition from a warm to cold ocean cavity. Tinto et al.[20] show that intrusion of warm mCDW is limited under the ice shelf front in the modern bed configuration, isolating WAIS from oceanic heat. But in a glacial maximum configuration, mCDW interacts with the ice sheet at the continental shelf edge. The Siple Coast's graben-bounding faults have been argued to accommodate the glacioisostatic rebound following the last deglaciation[71]. As the bed rebounded, basal melt rates proximal to WIS became influenced by cool HSSW produced by polynyas on one side of the tectonic boundary, whereas basal melt rates proximal to BIS and KIS became influenced by low-salinity ice shelf water on the other. The ISM indicates that WIS and MIS are more sensitive to changes in ocean forcing than BIS, but another factor contributing to the lag in BIS advance could be these differences in shelf water mass composition, as well as the timing of when these two sides of the cavity become isolated from mCDW. Our simulations do not rule out that bed topographic changes indirectly contributed to WAIS retreat and advance by altering the delivery of mCDW to the WAIS ice streams. Cavity-resolving regional ocean model simulations could test the relative influences of Late Holocene climatic conditions and bed topography on the sub-ice shelf ocean circulation and water mass properties.

In the Ross Embayment, WAIS has had a positive mass balance over the past decades[15], in its current state as a cold ocean cavity influenced by local polynyas[17]. However, our ISM simulations demonstrate that if conditions in the ocean cavity change and basal ice shelf melting increases, WAIS ice streams in this sector are susceptible to grounding line retreat. This implies that despite the relatively stable climate during the Holocene, extensive and rapid climate-driven grounding line migration has occurred in the past thousands of years. As Southern Ocean overturning and Antarctic sea ice decrease in response to human emissions of greenhouse gases[72,73], the Siple Coast

grounding line is vulnerable to an increase in ocean temperature in the Ross ice shelf cavity.

## Methods

### Ice sheet modeling

We employ the same ISM setup and approach as in ref. 30, but with a newer version of the Parallel Ice Sheet Model (version 2.0.3). The Parallel Ice Sheet Model is a three-dimensional thermomechanical ISM with a hybrid stress balance that combines shallow approximations of the flow equations for grounded and floating ice[28]. The model is run at 20 km horizontal resolution, but makes use of a subgrid grounding line scheme that is capable of representing grounding line reversibility at coarse resolution and compares well to higher-order models[74]. Iceberg calving is calculated from horizontal strain rates (i.e., faster calving in areas of fast flow[75]) and a heuristic minimum-thickness criterion to ensure that thin floating ice is removed.

We initiate our ISM experiments with a long spin-up procedure, the purpose of which is to allow the thermal structure of the ice sheets to evolve over a sufficiently long period that even deep layers are influenced by changing atmospheric conditions. We begin with an initial 20-yr smoothing in which only the Shallow Ice Approximation is used. Then, holding ice geometry fixed, we perform a 130-kyr thermal equilibration run using paleoclimate temperature forcing from the EPICA Dome C ice core (Jouzel et al.[76]), applied as an anomaly to the modern climatology[36]. We then perform a 130-kyr run in which we implement full model physics. Lastly, to ensure that our initial model state accurately reproduces glacial conditions at the start of the transient deglacial simulations, we precede each simulation with a 20 kyr period during which a constant "glacial maximum" climate field (i.e., 20 ka BP) is applied.

Our deglacial experiments are then run using surface climate forcing (surface temperature and precipitation) from the WAIS Divide ice core record (Fudge et al.[77]) applied as a percentage anomaly to a modern climatology[36], and ocean forcing from the Community Climate System Model 3 (CCSM3) TraCE-21ka experiment[37,38], with forcing updated every 100 years. We use a positive-degree day (PDD) model to translate temperatures above freezing into surface melt, of which 60% remains in the snowpack as a consequence of refreezing during percolation. Ice shelf basal melting is calculated from the TraCE-21ka ocean temperature and salinity fields at 500 m depth using a thermodynamic ocean model[78]. This parameterization is optimized with a melt factor to simultaneously reproduce present-day (1979–2010) melt rates in cold ocean cavities, such as the Ross ice shelf, as well as high melt areas, like Thwaites and Pine Island Glaciers[79,80]. We also apply a sea level forcing to the ice sheet based on a compilation of global sea level proxy records from the last deglaciation[81–84].

Similar to ref. 85, we performed an ensemble of experiments ($n = 270$) to explore model sensitivity to changes in flow enhancement factors, basal substrate, and sliding parameters, calving approximations, glacioisostatic adjustment, and surface climate and ocean thermal forcing in order to improve the representation of deglacial grounding line retreat in the Ross Sea. The model parameter values used for the simulations presented in the main text are listed in Table S1. Details on ISM benchmarking for the last deglaciation and present-day climate are provided in the Supplementary Information.

The effect of GIA on grounding line migration is a particular focus of our study. The ISM uses a one-dimensional viscoelastic Earth deformation model based on the fast Fourier transform solution of refs. 32 and 33. This two-layer model approximates the upper mantle as a linearly viscous half-space overlain by an elastic plate lithosphere. In our ISM ensemble, we adjust three parameters of the Earth deformation model over a wider range than previous studies[30,34,40,86], to investigate their impact on the Siple Coast grounding line: (1) mantle viscosity (1e19–5e21 Pa s$^{-1}$), (2) lithosphere flexural rigidity (1e23–5e25 N m), and (3) mantle density (3300–4500 kg m$^{-3}$). These parameter ranges cover the large spatial variability and uncertainty in solid Earth rheological properties of the Ross Sea region[24,25,70]. We performed experiments with different combinations of a weak vs. strong mantle and lithosphere to determine the effect of these parameters on the timing of WAIS grounding line retreat/readvance in the Ross embayment. Only a relatively narrow range of mantle viscosities (i.e., 5e20–1e21 Pa s) allows for this retreat/readvance to occur during the Holocene (Supplementary Fig. S2), though notably none of these experiments show retreat as extensive as indicated by the WAIS till samples[11].

The other focus of our study is uncertainty in ocean thermal forcing during the Holocene. The CCSM3 ocean component is relatively coarse resolution (nominal 3° horizontal resolution; 25 vertical levels) and does not resolve the ice shelf cavity. Modern basal melt rates of the Ross ice shelf are influenced by the Ross Sea and Terra Nova Bay polynyas[17], but CCSM3 does not simulate such processes. To consider the impact of past changes in local oceanographic conditions and their effect on ice shelf basal melting (e.g., ref. 21), we conducted experiments in which we modified the TraCE-21ka forcing by increasing Ross Sea temperature by different magnitudes (up to +0.6 °C) and for different lengths of time during the mid-to-late Holocene; i.e., the temperature increase occurs between 7 and 1 ka. This time period is chosen for the ocean forcing experiments because it coincides with the ice sheet retreat phase reconstructed in refs. 8 and 9 and modeled in ref. 10. Once the anomalous warming ceases, the ocean forcing is again derived directly from TraCE-21ka. We run these experiments only within the mantle viscosity range of 1e20–1e21 Pa s, lithosphere flexural rigidity range of 1e24–1e25 N m, and a mantle density of 3300 kg m$^{-3}$.

### Global glacioisostatic adjustment modeling

For modeling GRD effects associated with ice sheet advance and retreat, we use a 1D GIA model that solves the gravitationally self-consistent sea-level theory with a pseudo-spectral algorithm[26,87]. In addition, the sea-level theory we incorporate also allows shoreline migration and inundation of ocean area previously occupied by marine-based ice sheets. Viscoelastic deformation is solved on a self-gravitating Maxwell viscoelastic Earth with radially varying rheological structure represented by lithosphere thickness, upper and lower mantle viscosities, and the elastic and density structure is adopted from the seismic Preliminary Reference Earth Model (PREM)[88]. We note here that viscous deformation in the GIA model is solved based on the normal-mode theory in which each wavelength of relaxation has a distinct timescale.

We perform two sets of GIA ensemble simulations with different Earth structures. One set incorporates a thinner lithosphere thickness of 60 km with lower upper mantle viscosities of 1e18–1e19 Pa s and lower mantle viscosity of 1e22 Pa s (GIAWE), which best-fits GPS-measured solid Earth uplift in the Amundsen Sea sector of West Antarctica. The other set incorporates a thicker lithosphere thickness of 90 km upper mantle viscosity of 5e20 Pa s and the lower mantle viscosity of 5e21 Pa s (GIASE). For all GIA simulations in this study, we utilize the time window algorithm developed in ref. 27 that allows assigning of variable time resolution across the simulations, which makes glacial timescale simulations computationally more efficient. Using the time window algorithm, the global GIA model is updated with the new ice thickness history at every 100-year interval. By the end of the GIA simulation at 0 ka BP, the temporal resolution (i.e., the interval at which the GIA model reads in the ice history files) across the 40-ka long simulations is 1000 yr, 500 yr, 200 yr, and 100 yr for the time period 40–20 ka BP, 20–10 ka BP, 10–5 ka BP, and 5–0 ka BP, respectively.

The input Antarctic Ice Sheet history to the global GIA model is directly taken from the outputs of the standalone ISM simulations described above, specifically those shown in Fig. 2 as well as a few additional ISM simulations with higher lithosphere flexural rigidity

values ($n = 24$). We keep the Greenland Ice Sheet static throughout the GIA simulation; this is justifiable as the mass fluctuation in the Greenland Ice Sheet was significantly smaller compared to the AIS. As for the spatial resolution, we perform the calculations up to degree and order 512 of spherical harmonics. The initial topography for the Antarctic domain is taken from the ISM simulations and combined with the ETOPO2 for outside the domain defined by the ISM. To explore the impact of GIA on late Holocene ice sheet readvance shown in Fig. 6, we adopt the iterative coupling procedure in ref. 31 by which the ice history from the ISM over the last 40 ka is used as input to a GIA model simulation, and the resulting changes in bedrock elevation and bathymetry are passed back to the ISM to perform another simulation over the last 2 ka, repeating the process two times until near convergence is reached.

## Data availability
Model outputs are publicly accessible from the Open Science Framework: https://osf.io/xqs8w/.

## Code availability
PISM is freely available as open-source code from https://github.com/pism/pism.git. The GIA model code is freely available as open-source code from https://github.com/hollyhan/1DSeaLevelModel_FWTW.

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

## Acknowledgements

D.P.L., N.R.G., K.M.J., and R.M.M. were supported by the New Zealand Ministry for Business, Innovation, and Employment under contract ANTA1801 (Antarctic Science Platform) and contract C05X1702 (Global Change through Time). We gratefully acknowledge the developers of the Parallel Ice Sheet Model (PISM). The development of PISM was supported by NASA grant NNX17AG65G and NSF grants PLR-1603799 and PLR-1644277. We thank F. He and others involved in TraCE-21ka for climate model outputs and advice, as well as A. Malyarenko, S. Jendersie, and G. Dunbar for helpful discussions.

## Author contributions

D.P.L. developed the study concept and performed the ice sheet modeling and analysis. H.K.H. performed the global glacioisostatic adjustment modeling and analysis. N.R.G. contributed to the ice sheet model experimental setup and analysis. N.G. contributed to the glacioisostatic adjustment model experimental setup. K.M.J. and R.M.M. contributed to the paleoclimate proxy interpretation. D.P.L., H.K.H., N.R.G., N.G., K.M.J., and R.M.M. contributed to the interpretation of results and writing of the manuscript.

## Competing interests

The authors declare no competing interests.
