## [Peer Review File · Nature Communications]

Ocean cavity regime shift reversed West Antarctic grounding line retreat in the late HoloceneREVIEWER COMMENTS

Reviewer #1 (Remarks to the Author):

In the submitted manuscript entitled, "Ocean cavity regime shift reversed West Antarctic grounding line retreat in the late Holocene" Lowry and colleagues investigate the role of ocean thermal state in forcing grounding line re-advance following the last deglaciation. This work is timely, as the topic of grounding line retreat and re-advance in West Antarctica (and some parts of East Antarctica) has gained popularity since the first publication on the topic in 2015, even being highlighted in the latest IPCC report. The major findings of this work highlight that a switch in ocean cavity regime (from warm to cold) can re-advance grounding lines during the Holocene, specifically along the Siple Coast in the Ross Sea Embayment. While I cannot deny the novelty and importance of this work, I encountered notable flaws in the framing of the study and discussion of geologic timeframe that must be considered before this work is suitable for publication. In light of these comments, I provide major comments associated with those themes below.

1. The first paragraph of the paper's introduction is imprecise and does not accurately represent work on the topic or across the broader field of study.

First, the quoted sea-level equivalent ice volume is outdated (citing Fretwell et al., 2013) and inaccurate (4.3 m). The BedMachine dataset (Morlighem et al., 2020) was more recently used to generate a higher-precision estimate of ice thickness, that ultimately resulted in a refined sea-level equivalent ice volume of 5.3 +/- 0.2 m in West Antarctica. These volumes can be found in supplementary table 3, and the basins from which they are generated in supplementary table 2.

Second, the representation of previous efforts to date the grounding line retreat and re-advance do not represent the most recent work on the topic or the difference between a modeled constraint and a measured constraint. For the former, I recommend the author review recent work by Venturelli et al. (2023; AGU Advances) which is notably excluded from the citations in this manuscript. For the latter, more nuance should be added to the introduction to differentiate between the measured results (Venturelli et al., 2020; 2023) which provide a tight constraint on the timing (~7-6 ka) directly at the Whillans Grounding Zone and far upstream beneath Mercer Ice Stream and modeled results (Neuhaus et al., 2021) that represent a wider regional area, but notably have a high uncertainty due to the nature of the model.

Third, the introduction of the concept of grounding line retreat and re-advance does not accurately represent the work on the topic. Though the paper by Kingslake and colleagues (2018; Nature) popularized the idea that the process of grounding line retreat and re-advance could challenge the paradigm of monotonic retreat, it was not the first paper to discuss the topic or even the role of GIA in modulating the process. Bradley and colleagues (2015; EPSL) demonstrated this process in the Weddell Sea Embayment long before the Kingslake paper was published.

Taken individually, each of these points may seem a minor issue. However, their collective impact of multiple inaccuracies in the very introduction of the manuscript sets the manuscript off on the wrong foot. It should be a priority to represent the most up-to-date science in the field in the manuscript's introduction to properly set the stage for the study.

2. Though the authors do a notable job of comparing their results to proxy records of freshwater input and ocean temperature, comparison to on-ice cosmogenic nuclide exposure age arrays are not as comprehensive.

Perhaps the most well-studied area with respect to LGM-to-present exposure age records exists in the Transantarctic Mountains, with several studies here published in the last few years (e.g., Stutz et al., 2023; Hillebrand et al., 2021) yet only two studies (Todd et al., 2010; Spector et al., 2017) are

compared with results from the submitted work. The manuscript would be more robust if the authors considered the full suite of work on this topic and in this region. One useful resource could be the informal cosmogenic-nuclide exposure-age database (ICE-D; <https://ice-d.apps.pgc.umn.edu/antarctica/>) which demonstrates the data coverage in the area. I recognize that approaching data in a database such as this might be intimidating, but another possible avenue for comparison might be the ANTICE2 data compilation by Lecavalier et al. (2023; Earth System Science Data) which contextualizes much of the data presented in ICE-D. One final resource on this topic, in the context of grounding line retreat and re-advance in the area is a review paper that compiled these exposure age arrays (Johnson et al., 2022; The Cryosphere). Without a more thorough analysis of these data in the context of the modeling results in this study, the citations appear random and cherry picked.

3. At present, the submitted study discusses what forces retreat at WIS, KIS, and BIS, however geologic evidence from Mercer Ice Stream (MIS) also exists, and the exclusion of those data mean that the study does not robustly assess all available geologic constraints in the area. Given that the available geologic constraints for retreat and re-advance along the Siple Coast are few (N=2 studies, excluding constraints derived from models), it is important to consider all published work in the manuscript under review.

Venturelli and colleagues (2023) demonstrated that grounding line retreat at Mercer Ice Stream occurred far inland (~250 km) around ~6 ka, which is notably very similar to the dates presented at Whillans Grounding Zone (Venturelli et al., 2020). These new constraints present an opportunity to expand the work herein to a bound along the Siple Coast nearer the Transantarctic Mountains, but also presents compelling evidence that the timing over which retreat occurred (referred to as "retreat phase" in the submitted manuscript) was tighter than what was presented by Neuhaus and colleagues (2021).

Along these lines, the authors conflate model results (Neuhaus et al., 2021) with geologic evidence. Of course, it is important to consider the results of the submitted work within the context of those published model results, but using a modeled timing as a target would fundamentally be associated with a higher uncertainty in the result of Lowry and colleagues submitted work. A recent review paper by Johnson and colleagues (2022) provided a valuable discussion on the difference between direct and indirect evidence of grounding line retreat and re-advance that should be considered in the context of this comment.

To make broad conclusions about the retreat and re-advance of the entire Siple Coast, the authors should (a) include all studies along the Siple Coast and (b) consider the difference between a geologic constraint and a modeled result. Doing so will strengthen the validity of the conclusions herein.

Finally, it feels necessary to mention that the submitted work is novel and important. The consideration of both GIA and climate as forcing mechanisms of grounding line retreat and re-advance will enable a significant advance in our knowledge of the reversibility of retreat. I believe that consideration of the above points has the potential to greatly improve the chance that this manuscript impacts the field the way it should.

References cited in this review not included in the original manuscript:

Bradley, Sarah L., et al. "Low post-glacial rebound rates in the Weddell Sea due to Late Holocene ice-sheet readvance." *Earth and Planetary Science Letters* 413 (2015): 79-89.

Hillebrand, Trevor R., et al. "Holocene thinning of Darwin and Hatherton glaciers, Antarctica, and implications for grounding-line retreat in the Ross Sea." *The Cryosphere* 15.7 (2021): 3329-3354.

Johnson, Joanne S., et al. "Existing and potential evidence for Holocene grounding line retreat and readvance in Antarctica." *The Cryosphere* 16.5 (2022): 1543-1562.

Lecavalier, Benoit S., et al. "Antarctic ice sheet paleo-constraint database." *Earth System Science Data Discussions* 2022 (2022): 1-34.

Morlighem, Mathieu, et al. "Deep glacial troughs and stabilizing ridges unveiled beneath the margins of the Antarctic ice sheet." *Nature Geoscience* 13.2 (2020): 132-137.

Stutz, Jamey, et al. "Inland thinning of Byrd Glacier, Antarctica, during Ross Ice Shelf formation." *Earth Surface Processes and Landforms* (2023).

Venturelli, Ryan A., et al. "Constraints on the timing and extent of deglacial grounding line retreat in West Antarctica." *AGU Advances* 4.2 (2023): e2022AV000846.

Reviewer #2 (Remarks to the Author):

Lowry et al investigate the potential drivers for a Siple Coast grounding line retreat and re-advance during the late Holocene. Geological evidence suggests that the Siple Coast grounding line might have retreated substantially during the late Holocene and e.g. Kingslake et al. (2018) provide modelling and proxy indications that isostatic rebound was instrumental in driving grounding line re-advance subsequently to a pronounced episode of retreat past the present-day grounding line position. Venturelli et al. (2020) and Neuhaus et al. (2021) provided new age constraints for marine exposure of this sector, additionally Neuhaus et al. suggest that the re-advance of the grounding line was triggered by subtle regional climatic changes (reduced ocean thermal forcing). Lowry et al. explore an ensemble of ice sheet model simulations combining different scenarios of ocean thermal forcing combined with an integrated approach of GIA modelling and the 3D ice sheet model PISM. They test whether their model results can be reconciled with the timing/pacing of retreat and re-advance suggested by proxy reconstructions.

Lowry et al. show that modelled grounding line retreat was driven by an episodic mCDW intrusion into the ice shelf cavity leading to up to 0.5 K ocean warming (larger warming leads to full WAIS collapse and can therefore be ruled out). Subsequently re-advance was mainly driven by a combination between GIA response to ice load changes and transient cooling of the Ross ice shelf cavity driven by strong overturning and cutoff of mCDW. This is basically a combination of factors which have been suggested previously e.g. in Kingslake et al. 2018 and Neuhaus et al. 2021 the novelty being the application of a GIA model and thus a more realistic bedrock and sea level response to ice load changes.

The paper is well written and clearly structured. The model experiments discussed are highly relevant as they show that already subtle changes in the thermal regime of the Ross Ice Shelf cavity combined with GIA can lead to large fluctuations in the grounding line position.

This could be a valuable addition and while I think it would be better placed in more specialized journals it would certainly be suitable for Nature Communications as well. However, I do have some major comments which should be addressed before considering publication in Nature Communications. First and foremost, the authors should clearly communicate the biases and uncertainties associated with their modelling approach. This can be achieved via the following three additions (Supplementary Material).

1. The results of their model ensemble (N=270) in terms of ice volume, SLE volume, ice area etc are

not shown. There should be at least one figure illustrating these integrated diagnostics as a timeseries covering the LGM to today. The individual simulations which have been highlighted in the main manuscript should be highlighted in this timeseries as well.

2. The authors should provide 2D plots of ice thickness anomalies with respect to present day observations (e.g. BedMachine or BEDMAP2) for the simulations discussed in the paper. Likewise biases in surface velocities should be provided (compared to MeASURES). Simulated melt rates at present day should be illustrated compared to estimates from e.g. Adusumilli et al. (2020) or similar datasets. I see that in S5 ice thickness anomalies are illustrated for one sim. But it is unclear whether these anomalies are with respect to pd sim or pd obs.

3. The authors employ a grid resolution of 20 km which is rather coarse (see comments below). There should be a discussion how higher resolution (e.g. 10 km) would affect the timing of retreat and re-advance (in the main paper). Ideally the authors could provide simulations at 10 km for selected simulations such as MV7.5e20 + 0.5 for GIAWE and GIASE as well as LC. Kingslake et al. (2018) employ resolutions of 15, 10 and 7 km and find substantial impact on the temporal and spatial evolution of the grounding line. If a complete (40ka) run is computationally too expensive, the authors could restart at higher res at e.g. 20ka BP.

I am aware that the authors refer to previous publications e.g. Golledge et al. (2021) with respect to the ISM settings and uncertainties. Golledge et al. (2021) in turn refers to Clark et al. (2020) for ice sheet model settings. Clark et al use PISMv0.71, the authors employ PISMv2.1.1 which is one of the latest versions of the model. In the previous publications model uncertainties are not discussed in detail and the pivot from v0.7 to v2.1 will certainly have affected the ice sheet response to the parameterization employed. I noted that the `sia_e` and `q` parameters (shallow ice enhancement factor and pseudo plastic parameter in PISM) given in Clark et al. favor a very dynamic ice sheet i.e. quick deformation, fast flow. I do wonder how such a parameter combination plays out for present day or pre-industrial climate conditions. The authors do not provide PD reference simulations (neither in this publication nor in Golledge et al., 2021 or Clark et al. 2020). Such reference simulations would provide an idea how the ice sheet behaves under current conditions. Looking further back, Clark et al. reference Golledge et al. 2015 in their Method section for PISM. Golledge et al. 2015 provide an Extended data figure which compares surface elevation, grounding line position, surface elevation and velocity with BEDMAP2 & MeASURES.

They also mention that sea level equivalent ice volume in their spinup run is within 4% of the total SLE ice volume of Antarctica. Quoting here: "However, the positions of grounding lines and calving lines are well captured, and the sea-level-equivalent ice volume is within 4% of empirically calculated values".

This would amount to a mismatch of ca. 2.3 m for their present day spin-up which is considerable. They also don't mention whether the spin-up produces ice gain or loss (i.e. -2.3m or + 2.3 m). Using a completely new model version will change the ice sheet's evolution during the spin-up at least somewhat. Given the limited information on the model fit with respect to observations in the previous cited publications I would suggest providing a detailed description of the model spin-up and discussion of the result of the spin-up in form of SLE ice volume change, thickness, surface velocity RMSE (incl. 2D plot) using PISMv2.1.1. This would provide the reader with the necessary information regarding the biases of the model setup.

Below I provide a range of minor and major comments:

The figures need to be reworked e.g. positioning of fig numbering/labels, colorbar labels incomplete or missing (same in supplementary material), colormaps are not ideal. Lineplots (Figure 3) very busy. I suggest to check whether colormaps are appropriate for color blindness. Figures are poor resolution.

I am missing a figure on the de-glacial thickness evolution (ice thickness anomalies with respect to PD

obs. or PD sim.) of the ice sheet, this could be e.g. selecting the MV7.5e20 simulation which you focus on e.g. in Figure 4 and the section Gravitational, rotational and deformational (GRD) effects in the Ross Sea region.

This figure could ideally show the LGM ice extent and thickness anomaly with respect to PD and likewise for the 0ka BP time slice of the simulation.

L23 ... is sensitive to Holocene climate ...

Figure 1: add legend for dots.

L65/66 unclear how the spatial surface forcing is devised based on the WAIS divide ice core. Climate index or some kind of parameterization?

Ocean forcing is derived from a CCSM3 transient simulation, respective publication in 2009. I therefore suggest a short discussion in the main part of the manuscript of the known biases and limitations. Additionally, it would be interesting to hear the opinion of the authors on the impact of updating the climate forcing every 100 years (i.e. in contrast to annual updates).

Ultimately the authors employ two different approaches for the forcing

1. Ice core-based climate reconstructions + GCM ocean output.
2. The same but with synthetic ocean temperature anomalies.

Figure 2. Suggest adding the PD observed grounding line so the reader gets a bearing as to where the simulated grl. ends up in respect to the observed one. Additionally in panel (I) there seem to be either two 0ka grounding lines or it shows an almost fully collapsed WAIS.

I suggest changing the topography colormaps in the figures (e.g. to a grayscale) and plot the grounding line evolution in distinguishable colors.

The authors run their simulations on 20 km resolution, the lateral extent of the ice streams populating the Ross sector of the WAIS thus varies between ca. 1-2 grid boxes. This goes back to my previous comment on providing the modelled output at present day compared to observations. In this case, specifically the modelled surface velocity and thickness in the Ross Sector compared to MeASURES and BedMACHINE (e.g. providing RMSE of the model ensemble and 2D vel.-difference plot for the best fitting ensemble member).

L88 how is an ice stream location defined in the proxy records and how do you compare it to the modelled location (midpoint of a central streamline?). Please elaborate.

L103: as shown in Kingslake et al. 2018.

Figure 3: What is the motivation to include the red line (simulations with +0.6C temperature anomaly)? This specific simulation suggests a collapsed WAIS at present day and thus should be considered unrealistic. I suggest removing it from the plot which would improve readability (lots of lines, colors, linestyles already). Also, where are the horizontal dashed lines? I don't see any. What is the meaning of the gray line (position of WIS and BIS). Is that the inferred paleo grounding zone? And if so, at which time? What are the colored vertical lines indicating? In general, I find this figure difficult to entangle. How do you define the grounding line position over time? Do you take a transient or constant central flowline and pinpoint the grounding line position along this flowline?

Reference list: references are all small caps except first letter. Maybe an issue with the export from the respective bibliography-software.

L111: ... has contributed to variable sea ...

L114: If I understand correctly the PISM simulations are used as an input for the GIA model. However, in the PISM output the Lingle-Clark bedrock deformation model is used so some kind of bedrock response to changing ice load is already included. This way you force GIA with data which already contains a bedrock response (albeit a less realistic one). I find this difficult to disentangle. But maybe I misunderstand something here.

Figure 4: missing label for colorbar. I do not understand how bed topg. changes between 2-7 ka BP (rather 7-2ka BP) are illustrated. Is this a mean topg-change during this period? Or the bedrock response at the end of the period (2ka BP)? Also, your notation is a little confusing. I assume you always implicitly mean BP but then I'd write 7 – 2 ka BP instead of 2 – 7 ka.

Supplementary figures: suggest checking the aspect ratio for all 2D plots. AIS looks a bit "squished". Also please add complete label, now only unit is given (m). All figures add PD observed grounding line for reference.

L127: which ice cover changes. From this line it seems you focus on one ISM realization. Which one? This should get a dedicated figure (see comment above).

L146 I don't find any mention of what uplift would be significant for sub-ice shelf ocean circulation in Tinto et al. This brings me back to Figure 4. If you write bed topography changes for 0-2ka (2 – 0 ka BP), do you mean a temporal average? Or is this depicting the topo changes at PD? If so, shouldn't they be very close to zero? Topo changes of 100-300 meters would point to an inconsistent ice load history, wouldn't they?

L151 effect on what? Response of what? Relative "cooling" or "warming"? Figure 3 shows that synthetic t-anomalies are all positive. Or do you refer to the standard forcing?

L152 isn't it the other way around? Ice sheet thins and retreats due to increasing basal melt rates? Or do you refer to increases in melt rates due to deeper grounding lines i.e. lower melting point?

L154 specify! Not everyone knows what the current melt rates in the Amundsen sea sector are. Also peak melt rates? Can be over 100 m/a underneath Thwaites or PIG, alternatively bulk rates (10-20 m/a).

Figure 5: for me it's rather difficult to identify which line color corresponds to what time. I suggest using a more variable cmap and including a legend. The overlapping bedrock and lower ice shelf boundary make this plot difficult to read. Again, poor resolution, suggest using vector graphics or raster with ≥ 300 dpi.

L161 yes, increased snowfall can increase discharge, but this discharge does not offset the SMB increase completely (about 30-60% according to Winkelmann et al., 2012). Also, this refers to SMB changes integrated over the entire continent. Regional increase in SMB e.g. at the grounding line wouldn't increase the dynamic ice flux necessarily because they might even reduce surface gradients and thus lower driving forces.

L163 which, in our simulation/can/ lead(s) to an instantaneous ... in the transition between the retreat and advance phase.

L164 Surface mass balance would require a 30-fold increase ... : increase with respect to what?

L165 how does that relate to the findings in Neuhaus et al., 2021 who suggest that regional climate change (modest ocean cooling) was the culprit behind grounding line re-advance?

Paragraph 174-188: the ice sheet response to the different bedrock scenarios discussed in the text are difficult to spot in Fig. 6 due to the choice of cmaps. I would suggest picking a more distinguishable colormap with perceivable color/contrast changes for the grounding line positions at different times. For the bedrock anomaly cmap you could consider picking white for the transition across zero (e.g. simple RdBu cmap).

Case c) in Fig. 6 suggests substantial uplift underneath the present day Ross ice shelf leading to ice rises/ice domes on islands. Wouldn't that be a strong indicator that the combination of ice load history and mantle viscosity is unrealistic and should be excluded? This should at least be mentioned (e.g. in line 179).

L195 picking up the point from above, doesn't the strong bed uplift on the Ross shelf leading to extensive ice rises/domes at present day disqualify a GIAWE structure?

L209 deuterium isotopes

L238-240 A perspective on how (un)realistic the GIAWE results are in terms of ice history and bed rebound is missing (see comments above).

L255 please provide parameters chosen for Eigencalving and thickness calving here. Even better, provide table of key parameters/forcings changed in the ensemble (i.e. flow enhancement factors, basal substrate, and sliding parameters, calving approximations, glacioisostatic, adjustment, and surface climate and ocean thermal forcing) e.g. in the supplements.

L262 please specify how you construct the surface climate forcing? WAIS divide record is a point measurement, how do you translate this to a spatial forcing covering the entire ice sheet domain?

L268-270 please provide a comparison (supplements) of simulated pd melt rates and estimates (e.g. Adumusilli et al. 2020.).

Reviewer #3 (Remarks to the Author):

Lowry et al., present a proxy-informed series of ice sheet model simulations to investigate if Holocene grounding line (GL) retreat in the Siple Coast region was driven by climatic (ocean, atmospheric forcing) or non-climatic processes (glacioisostatic adjustment). This is very timely paper, with significant implications for ice sheet stability. It is also the first paper to try and attribute forcing to Holocene GL retreat/overshoot. I can't comment on the specifics of the modelling, but the approach seems well conceived and thorough. Generally, the paper is very well written with appropriate/informative figures.

My only substantive comment relates to the timing of change(s), and how they are presented in the MS. I think the authors can do a much better job in highlighting the uncertainty associated with the 'timing' of retreat and readvance (derived from published literature). Currently there's a very large window when retreat is thought to have occurred (quoted as 7.5 to 2.0 ka in the MS, although I think this should be 7.2 - 1.7 ka based on the Venturelli and Neuhaus papers), with readvance more tightly constrained to after ~1.5 ka (note that in Figure 7, the pink interval/readvance phase is more like ~2.3 ka to present and this should be updated or better justified). Furthermore, all these ages, especially those in Neuhaus have very large uncertainty attached to them. On the upper end, GL retreat at Whillans grounding zone and Mercer occurred at ~7.2 and 6.3 ka, respectively (Venturelli et al., 2020; 2023). Conversely, retreat on Bindschadler and Kamb was as late as 1.8 ka and 1.7 ka (Neuhaus et al., 2021), with readvance as recently as 0.8 ka. This results in a very large target to work with! In this respect, I wonder if the MS would be stronger if you explicitly set this up as

sensitivity experiment rather than needing the model to replicate retreat/advance at a particular time? The large spread of GL retreat ages also creates problems relating to the forcing and how these model simulations fit with available proxy data/current thinking regarding ocean/atmosphere forcing i.e., it's very difficult to attribute a particular glacial response to a driver when the timing is so uncertain and variable across the Siple Coast. For e.g., Lowry et al suggest that a switch from a warm cavity to cold cavity can be used to explain retreat and re-advance, but proxy data imply that this transition occurred after 3.6 ka. Can this transition really explain GL retreat of Bindschadler and Kamb after 1.8 ka? Interestingly, Hall et al. (2006). PNAS., infer a warmer than present interval in the western Ross Sea ~2.3 to 1.1 14C kyr BP (Hall et al., 2006), which is consistent with the timing of retreat for Bindschadler and Kamb but not necessarily the 'cold cavity' scenario presented in the MS and summarised in Figure 7.

Thus, in summary, I think the revised MS needs to be much more precise/transparent regarding the timing of GL retreat – GL changes occurring at 7.2 ka are probably not driven by the same processes as changes at 1.7 ka. You could include this in the main body of the MS or as an extra section in the supplementary information. Given the large spread of GL retreat ages, I would also like to see more consideration about the factors driving the apparent spatial variability. Venturelli et al. (2020) speculated that differences in ocean forcing in the eastern and western Ross Sea might explain the discrepancy between their age (7.2 ka) and the earlier GL retreat age (9.7 ka) in Kingslake et al. (2018). Can you say anything more about this? Rather than try and say that warm/cold cavity transition explains all GL variability, I think it would be more instructive to set out which ice streams/ice sheet thinning histories do not fit this model?

Minor comments:

Lines 29-30: I don't think its necessary to specify 'modified' (-CDW) unless you want to discuss on shelf properties/oceanography.

Line 30: Walker et al is a great paper but an odd choice here. Jacobs et al., 1996. GRL; Jenkins et al., 2009. Nat Geo; Dutrieux et al., 2014. Science?

Lines 74-77: Bit of a cop out. Either the timing matters or it doesn't. Much of the discussion revolves around when retreat/readvance occurred and what drove these changes.

Line 96: Choice of +0.6 deg C simulation; if this leads to ice shelf collapse does this imply that your model is too sensitive to ocean forcing? I note in the methods that the 'high melt rate' parameter/end member is derived from Thwaites/PIG but is this forcing realistic for the Ross Sea during the Holocene?

Line 136: refs out of order. Ref 46 should be 47?

Lines 200-202: Re-word. Mid-Holocene peak temperatures don't explain retreat of KIS (1.8 ka) and BIS (1.7 ka), only WIS (4.7 ka). Also include age-data from Mercer.

Fig. 7. As previously noted, background colours (blue, yellow, pink), which correspond to developing, warm and cold cavities are too simplistic to explain variable timing of GL retreat? One implication could be that BIS and WIS are behaving differently from WIS/Whillans grounding zone and Mercer? Would be more interesting to speculate about what drove this?

Lines 320 (refs): A couple of recent papers not cited but are relevant are: Jones et al., 2023 (<https://agupubs.onlinelibrary.wiley.com/doi/full/10.1029/2020GL091454>) also suggests enhanced ocean-driven melt in the Early-to-Mid Holocene.

Melis et al. (<https://jm.copernicus.org/articles/40/15/2021/jm-40-15-2021.pdf>) indicate stronger bottom currents in late Holocene.

Venturelli et al. (2023) (<https://agupubs.onlinelibrary.wiley.com/doi/epdf/10.1029/2022AV000846>) provide constraints on the timing of GL retreat Mercer Ice Stream.

We thank all three Reviewers for insightful comments that have greatly improved the manuscript. Our responses to individual comments are below in blue.

REVIEWER COMMENTS

Reviewer #1 (Remarks to the Author):

In the submitted manuscript entitled, “Ocean cavity regime shift reversed West Antarctic grounding line retreat in the late Holocene” Lowry and colleagues investigate the role of ocean thermal state in forcing grounding line re-advance following the last deglaciation. This work is timely, as the topic of grounding line retreat and re-advance in West Antarctica (and some parts of East Antarctica) has gained popularity since the first publication on the topic in 2015, even being highlighted in the latest IPCC report. The major findings of this work highlight that a switch in ocean cavity regime (from warm to cold) can re-advance grounding lines during the Holocene, specifically along the Siple Coast in the Ross Sea Embayment. While I cannot deny the novelty and importance of this work, I encountered notable flaws in the framing of the study and discussion of geologic timeframe that must be considered before this work is suitable for publication. In light of these comments, I provide major comments associated with those themes below.

We appreciate Reviewer 1 for noting the importance of the study topic and the novelty of our approach, as well as their constructive comments regarding the wider geological context. Our responses to specific comments are below.

1. The first paragraph of the paper’s introduction is imprecise and does not accurately represent work on the topic or across the broader field of study.

First, the quoted sea-level equivalent ice volume is outdated (citing Fretwell et al., 2013) and inaccurate (4.3 m). The BedMachine dataset (Morlighem et al., 2020) was more recently used to generate a higher-precision estimate of ice thickness, that ultimately resulted in a refined sea-level equivalent ice volume of 5.3 +/- 0.2 m in West Antarctica. These volumes can be found in supplementary table 3, and the basins from which they are generated in supplementary table 2.

This has been corrected.

Second, the representation of previous efforts to date the grounding line retreat and re-advance do not represent the most recent work on the topic or the difference between a modeled constraint and a measured constraint. For the former, I recommend the author review recent work by Venturelli et al. (2023; AGU Advances) which is notably excluded from the citations in this manuscript. For the latter, more nuance should be added to the introduction to differentiate between the measured results (Venturelli et al., 2020; 2023) which provide a tight constraint on the timing (~7-6 ka) directly at the Whillans Grounding Zone and far upstream beneath Mercer Ice Stream and modeled results (Neuhaus et al., 2021) that represent a wider regional area, but notably have a high uncertainty due to the nature of the model.

We thank the Reviewer for alerting us to the recent work of Venturelli et al. (2023), which was published while we were drafting the manuscript and was unfortunately missed. The additional geologic constraint from Mercer Ice Stream is now incorporated in our revised figures, as well as discussed throughout the manuscript. We have also clarified the methods employed by the various studies in the Introduction and Results, that is, radiocarbon dating of the subglacial material in Venturelli et al. (2020) and (2023) using the ramped pyrolysis-oxidation ¹⁴C method versus

radiocarbon input and decay modelling in Neuhuas et al., (2021). We specify the uncertainty ranges of both methods from the respective studies.

Lines 18-22 (Introduction):

“In the Ross Embayment (Fig 1), radiocarbon dating of subglacial till from the Whillans and Mercer ice streams (WIS and MIS, respectively) has indicated WAIS retreated hundreds of kilometres upstream of its current position between 7.5 and 5.3 ka BP^{14, 15}. The timing of WAIS readvance in this sector has been inferred from radiocarbon input and decay modelling to have occurred within the past 1.7 ka for the WIS, and even more recently for the Kamb and Bindschadler ice streams (KIS and BIS, respectively)¹⁶.”

Lines 109-120 (Results: Ice sheet response to ocean forcing):

“Radiocarbon dating based on ramped pyrolysis of low-carbon grounding-line-proximal sediments^{7, 8} and input and decay modelling⁹ offer age constraints for marine exposure of the ice stream locations which are used here to assess the model simulations (Fig 3). At Whillans Grounding Zone (WGZ), close to the modern grounding line, Venturelli et al. (2020)⁷ suggests a mid-Holocene grounding line retreat occurring at 7.2 ka BP, with an uncertainty range from 7.5 to 4.8 ka BP. Simulations with mantle viscosity of 5e20 and 7.5e20 Pa s show a consistent timing of retreat at this site with an ocean temperature anomaly of at least +0.4°C. Even without any imposed Holocene ocean warming, simulations using a mantle viscosity of 1e21 Pa s show retreat that is >1.5ka too early. 250 km further upstream at the MIS site (see Fig 1), Venturelli et al. (2023)⁸ constrain retreat to 6.3 ka BP ± 1 ka, indicating the retreat may have been more rapid than the simulations shown in Fig 3. More rapid grounding line retreat of these southern ice streams is possible with either earlier onset of anomalous ocean warming, or with higher temperatures (Fig S6); though in these simulations, ice shelf collapse occurs if ocean temperatures exceed +0.5°C for >1000 ka, hence we consider +0.5°C to be the upper limit of plausibility.”

Lines 217-219 (Discussion):

“At Taylor Dome, MSA is low in the early Holocene, consistent with widespread sea ice cover and less open ocean area. In the middle Holocene, from 7.1 ka BP, land fast sea ice declines in the Western Ross Sea, with peak ocean temperatures at 5.2 ka BP⁴⁵, coinciding with retreat of WIS and MIS^{7, 8}.”

Third, the introduction of the concept of grounding line retreat and re-advance does not accurately represent the work on the topic. Though the paper by Kingslake and colleagues (2018; Nature) popularized the idea that the process of grounding line retreat and re-advance could challenge the paradigm of monotonic retreat, it was not the first paper to discuss the topic or even the role of GIA in modulating the process. Bradley and colleagues (2015; EPSL) demonstrated this process in the Weddell Sea Embayment long before the Kinglake paper was published.

Given the regional focus of our study to the Ross Sea region, we neglected to cite Bradley et al. (2015) in the previous version of the manuscript. We agree with the Reviewer that this study is important for the wider geologic context of West Antarctic Ice Sheet dynamics during the Holocene, particularly with respect to the hypothesis of isostatic rebound as a driver of ice sheet advance. This citation has been added in this revised version:

Lines 37-41 (Introduction):

An alternative explanation for WAIS retreat and advance was first suggested by Bradley et al. (2015)⁴ for the Weddell Embayment, which noted that low modern bed uplift rates could be explained by extensive grounding line retreat during the last deglaciation when the bed was lower, with subsequent isostatic rebound causing regrounding of the ice sheet. Kingslake et al. (2018)¹⁰ further explored this mechanism for the wider WAIS, including the Ross Embayment, using process-based ice sheet model simulations.”

Taken individually, each of these points may seem a minor issue. However, their collective impact of multiple inaccuracies in the very introduction of the manuscript sets the manuscript off on the wrong foot. It should be a priority to represent the most up-to-date science in the field in the manuscript’s introduction to properly set the stage for the study.

We understand the Reviewer’s concerns about the introduction and have addressed these points accordingly.

2. Though the authors do a notable job of comparing their results to proxy records of freshwater input and ocean temperature, comparison to on-ice cosmogenic nuclide exposure age arrays are not as comprehensive.

Perhaps the most well-studied area with respect to LGM-to-present exposure age records exists in the Transantarctic Mountains, with several studies here published in the last few years (e.g., Stutz et al., 2023; Hillebrand et al., 2021) yet only two studies (Todd et al., 2010; Spector et al., 2017) are compared with results from the submitted work. The manuscript would be more robust if the authors considered the full suite of work on this topic and in this region. One useful resource could be the informal cosmogenic-nuclide exposure-age database (ICE-D; <https://ice-d.apps.pgc.umn.edu/antarctica/>) which demonstrates the data coverage in the area. I recognize that approaching data in a database such as this might be intimidating, but another possible avenue for comparison might be the ANTICE2 data compilation by Lecavalier et al. (2023; Earth System Science Data) which contextualizes much of the data presented in ICE-D. One final resource on this topic, in the context of grounding line retreat and re-advance in the area is a review paper that compiled these exposure age arrays (Johnson et al., 2022; The Cryosphere). Without a more thorough analysis of these data in the context of the modeling results in this study, the citations appear random and cherry picked.

We focused on surface exposure records in closest proximity to the Siple Coast, but have added a new figure to the Supplemental Information (Fig S2) that compares our ice thinning model predictions with the more recently published records from Hillebrand et al. (2021) and Stutz et al., (2023):

Fig S2, which shows deglacial ice volume and ice thickness evolution of our full model ensemble (gray lines) and reference model (colored), with comparison to ice thinning records from the Transantarctic Mountains:

3. At present, the submitted study discusses what forces retreat at WIS, KIS, and BIS, however geologic evidence from Mercer Ice Stream (MIS) also exists, and the exclusion of those data mean that the study does not robustly assess all available geologic constraints in the area. Given that the available geologic constraints for retreat and re-advance along the Siple Coast are few (N=2 studies, excluding constraints derived from models), it is important to consider all published work in the manuscript under review.

Venturelli and colleagues (2023) demonstrated that grounding line retreat at Mercer Ice Stream occurred far inland (~250 km) around ~6 ka, which is notably very similar to the dates presented at Whillans Grounding Zone (Venturelli et al., 2020). These new constraints present an opportunity to expand the work herein to a bound along the Siple Coast nearer the Transantarctic Mountains, but also presents compelling evidence that the timing over which retreat occurred (referred to as “retreat phase” in the submitted manuscript) was tighter than what was presented by Neuhaus and colleagues (2021).

We agree with the Reviewer that this new data is extremely valuable for model benchmarking and process understanding. We have added the location of the Mercer Ice Stream site in Figures, 1, 2, 4 and 6, as well as in the Supplemental Information Figures S4, S5 and S7. We also discuss our model results in context to the findings of Venturelli et al. (2023), and refer the Reviewer to the above comment with specific text additions in the revised manuscript.

Along these lines, the authors conflate model results (Neuhaus et al., 2021) with geologic

evidence. Of course, it is important to consider the results of the submitted work within the context of those published model results, but using a modeled timing as a target would fundamentally be associated with a higher uncertainty in the result of Lowry and colleagues submitted work. A recent review paper by Johnson and colleagues (2022) provided a valuable discussion on the difference between direct and indirect evidence of grounding line retreat and re-advance that should be considered in the context of this comment.

While a comparison between the methods of Neuhaus et al. (2021) to Venturelli et al. (2020; 2023) is beyond the scope of this study, we now clearly specify that two different methods are used; specifically that the Neuhaus et al. (2021) ages are derived from a model. Notably, the uncertainty ranges of Neuhaus et al. (2021) are larger, but they do show overlap with the age constraints of ice sheet retreat provided by Venturelli et al. (2020) and Venturelli et al. (2023). Neuhaus et al. (2021) offers the only available estimation for the timing of the ice sheet readvance of which we are aware.

To make broad conclusions about the retreat and re-advance of the entire Siple Coast, the authors should (a) include all studies along the Siple Coast and (b) consider the difference between a geologic constraint and a modeled result. Doing so will strengthen the validity of the conclusions herein.

Finally, it feels necessary to mention that the submitted work is novel and important. The consideration of both GIA and climate as forcing mechanisms of grounding line retreat and re-advance will enable a significant advance in our knowledge of the reversibility of retreat. I believe that consideration of the above points has the potential to greatly improve the chance that this manuscript impacts the field the way it should.

We thank the Reviewer for their suggestions on citations to add to improve the geologic context presented in our paper, and for noting the novelty of our modelling approach. We have added the references listed below to the revised manuscript.

References cited in this review not included in the original manuscript:

Bradley, Sarah L., et al. "Low post-glacial rebound rates in the Weddell Sea due to Late Holocene ice-sheet readvance." *Earth and Planetary Science Letters* 413 (2015): 79-89.

Hillebrand, Trevor R., et al. "Holocene thinning of Darwin and Hatherton glaciers, Antarctica, and implications for grounding-line retreat in the Ross Sea." *The Cryosphere* 15.7 (2021): 3329-3354.

Johnson, Joanne S., et al. "Existing and potential evidence for Holocene grounding line retreat and readvance in Antarctica." *The Cryosphere* 16.5 (2022): 1543-1562.

Lecavalier, Benoit S., et al. "Antarctic ice sheet paleo-constraint database." *Earth System Science Data Discussions* 2022 (2022): 1-34.

Morlighem, Mathieu, et al. "Deep glacial troughs and stabilizing ridges unveiled beneath the margins of the Antarctic ice sheet." *Nature Geoscience* 13.2 (2020): 132-137.

Stutz, Jamey, et al. "Inland thinning of Byrd Glacier, Antarctica, during Ross Ice Shelf formation." *Earth Surface Processes and Landforms* (2023).

Venturelli, Ryan A., et al. "Constraints on the timing and extent of deglacial grounding line retreat

in West Antarctica." AGU Advances 4.2 (2023): e2022AV000846.

Reviewer #2 (Remarks to the Author):

Lowry et al investigate the potential drivers for a Siple Coast grounding line retreat and re-advance during the late Holocene. Geological evidence suggests that the Siple Coast grounding line might have retreated substantially during the late Holocene and e.g. Kingslake et al. (2018) provide modelling and proxy indications that isostatic rebound was instrumental in driving grounding line re-advance subsequently to a pronounced episode of retreat past the present-day grounding line position. Venturelli et al. (2020) and Neuhaus et al. (2021) provided new age constraints for marine exposure of this sector, additionally Neuhaus et al. suggest that the re-advance of the grounding line was triggered by subtle regional climatic changes (reduced ocean thermal forcing). Lowry et al. explore an ensemble of ice sheet model simulations combining different scenarios of ocean thermal forcing combined with an integrated approach of GIA modelling and the 3D ice sheet model PISM. They test whether their model results can be reconciled with the timing/pacing of retreat and re-advance suggested by proxy reconstructions.

Lowry et al. show that modelled grounding line retreat was driven by an episodic mCDW intrusion into the ice shelf cavity leading to up to 0.5 K ocean warming (larger warming leads to full WAIS collapse and can therefore be ruled out). Subsequently re-advance was mainly driven by a combination between GIA response to ice load changes and transient cooling of the Ross ice shelf cavity driven by strong overturning and cutoff of mCDW. This is basically a combination of factors which have been suggested previously e.g. in Kingslake et al. 2018 and Neuhaus et al. 2021 the novelty being the application of a GIA model and thus a more realistic bedrock and sea level response to ice load changes.

The paper is well written and clearly structured. The model experiments discussed are highly relevant as they show that already subtle changes in the thermal regime of the Ross Ice Shelf cavity combined with GIA can lead to large fluctuations in the grounding line position.

This could be a valuable addition and while I think it would be better placed in more specialized journals it would certainly be suitable for Nature Communications as well. However, I do have some major comments which should be addressed before considering publication in Nature Communications. First and foremost, the authors should clearly communicate the biases and uncertainties associated with their modelling approach. This can be achieved via the following three additions (Supplementary Material).

We thank Reviewer 2 for their positive comments and their constructive suggestions to improve communication about the ice sheet model validation and figure presentation. Our responses to specific comments are below.

1. The results of their model ensemble (N=270) in terms of ice volume, SLE volume, ice area etc are not shown. There should be at least one figure illustrating these integrated diagnostics as a timeseries covering the LGM to today. The individual simulations which have been highlighted in the main manuscript should be highlighted in this timeseries as well.

Done. See Fig S2 in the Supplementary Information, and above as a response to Reviewer 1.

2. The authors should provide 2D plots of ice thickness anomalies with respect to present day observations (e.g. BedMachine or BEDMAP2) for the simulations discussed in the paper.

Likewise biases in surface velocities should be provided (compared to MeASUREs). Simulated melt rates at present day should be illustrated compared to estimates from e.g. Adusumilli et al. (2020) or similar datasets. I see that in S5 ice thickness anomalies are illustrated for one sim. But it is unclear whether these anomalies are with respect to pd sim or pd obs.

We have added a plot showing present-day ice thickness, surface velocity and ice shelf basal melt rates of the ice sheet model (Fig S3):

3. The authors employ a grid resolution of 20 km which is rather coarse (see comments below). There should be a discussion how higher resolution (e.g. 10 km) would affect the timing of retreat and re-advance (in the main paper). Ideally the authors could provide simulations at 10 km for selected simulations such as MV7.5e20 + 0.5 for GIAWE and GIASE as well as LC. Kingslake et al. (2018) employ resolutions of 15, 10 and 7 km and find substantial impact on the temporal and spatial evolution of the grounding line. If a complete (40ka) run is computationally too expensive, the authors could restart at higher res at e.g. 20ka BP.

Following the Reviewer's advice, we have run sensitivity experiments with a spatial resolution of 10km from 20ka BP. This increase in spatial resolution substantially increases model run time on our machine, which makes running the full ensemble at this resolution too computationally costly to do. Increasing spatial resolution results in faster ice sheet/shelf flow and thinner ice shelves. Accordingly, we reduced the SIA and SSA enhancement factors for these experiments. We have added the results of one of these 10km resolution experiments, which uses the 0.5°C+ ocean forcing, in Fig S6 to demonstrate that the ocean forcing mechanism for WAIS retreat and advance is robust and independent of model resolution. We did not re-run the GIA models with these 10km simulations because bedrock elevation changes due to GIA generally occur over larger spatial scales and our 20km resolution simulations are sufficient for capturing this effect.

I am aware that the authors refer to previous publications e.g. Golledge et al. (2021) with respect to the ISM settings and uncertainties. Golledge et al. (2021) in turn refers to Clark et al. (2020) for ice sheet model settings. Clark et al use PISMv0.71, the authors employ PISMv2.1.1 which is one of the latest versions of the model. In the previous publications model uncertainties are not discussed in detail and the pivot from v0.7 to v2.1 will certainly have affected the ice sheet response to the parameterization employed. I noted that the `sia_e` and `q` parameters (shallow ice enhancement factor and pseudo plastic parameter in PISM) given in Clark et al. favor a very dynamic ice sheet i.e. quick deformation, fast flow. I do wonder how such a parameter combination plays out for present day or pre-industrial climate conditions. The authors do not provide PD reference simulations (neither in this publication nor in Golledge et al., 2021 or Clark et al. 2020). Such reference simulations would provide an idea how the ice sheet behaves under current conditions. Looking further back, Clark et al. reference Golledge et al. 2015 in their Method section for PISM. Golledge et al. 2015 provide an Extended data figure which compares surface elevation, grounding line position, surface elevation and velocity with BEDMAP2 & MeASURES.

They also mention that sea level equivalent ice volume in their spinup run is within 4% of the total SLE ice volume of Antarctica. Quoting here: "However, the positions of grounding lines and calving lines are well captured, and the sea-level-equivalent ice volume is within 4% of empirically calculated values".

This would amount to a mismatch of ca. 2.3 m for their present day spin-up which is considerable. They also don't mention whether the spin-up produces ice gain or loss (i.e. -2.3m or + 2.3 m). Using a completely new model version will change the ice sheet's evolution during the spin-up at least somewhat. Given the limited information on the model fit with respect to observations in the previous cited publications I would suggest providing a detailed description of the model spin-up and discussion of the result of the spin-up in form of SLE ice volume change, thickness, surface velocity RMSE (incl. 2D plot) using PISMv2.1.1. This would provide the reader with the necessary information regarding the biases of the model setup.

More recent versions of the VUW-PISM model under present-day conditions are shown in Golledge et al. (2019) (PISM 0.7.3; See Fig 1 and Extended Data Fig 7), Seroussi et al., (2020) (PISM 0.7.3; see Fig 1 and 3), and Lowry et al. (2021) (PISM v1.1.2; see Fig S1 and S2). The RMSE for ice thickness and ice surface velocity are within the range of the ISMIP6-Antarctica models. Because these runs are performed with a coarser resolution and with a newer version of PISM we have included the present-day ice thickness, velocity, and melt rates in our revised Supplementary Information (see Fig S3), as suggested by the Reviewer.

Below I provide a range of minor and major comments:

The figures need to be reworked e.g. positioning of fig numbering/labels, colorbar labels

incomplete or missing (same in supplementary material), colormaps are not ideal. Lineplots (Figure 3) very busy. I suggest to check whether colormaps are appropriate for color blindness. Figures are poor resolution.

We have ensured that all figures in this revised manuscript are 300 dpi and use appropriate colormaps to accommodate color blindness. We thank the Reviewer for suggestions and respond about specific figures to comments below.

I am missing a figure on the de-glacial thickness evolution (ice thickness anomalies with respect to PD obs. or PD sim.) of the ice sheet, this could be e.g. selecting the MV7.5e20 simulation which you focus on e.g. in Figure 4 and the section Gravitational, rotational and deformational (GRD) effects in the Ross Sea region.

This figure could ideally show the LGM ice extent and thickness anomaly with respect to PD and likewise for the 0ka BP time slice of the simulation.

We have added Figure S2, which shows deglacial ice thinning changes predicted by the ice sheet model in comparison to surface exposure records from the Transantarctic Mountains. Revised Fig S4 and S5 now show the LGM grounding line of the various experiments. Fig S8 shows ice thickness changes during the Holocene.

L23 ... is sensitive to Holocene climate ...

Fixed.

Figure 1: add legend for dots.

The circles are labelled.

L65/66 unclear how the spatial surface forcing is devised based on the WAIS divide ice core. Climate index or some kind of parameterization?

The surface temperature and surface mass balance are applied as percentage anomalies relative to the present-day surface climatology from Van Wessem et al. (2014). This is now specified.

Ocean forcing is derived from a CCSM3 transient simulation, respective publication in 2009. I therefore suggest a short discussion in the main part of the manuscript of the known biases and limitations. Additionally, it would be interesting to hear the opinion of the authors on the impact of updating the climate forcing every 100 years (i.e. in contrast to annual updates).

We have added the following:

Lines 74 to 83 (Results: Ice sheet response to ocean forcing):

“While the surface climate evolution of Antarctica is relatively well known over the last deglaciation from the distribution of Antarctic ice core records, ocean circulation under ice shelves is poorly constrained. Different approaches exist for the implementation of ocean forcing in paleo-ice sheet simulations, including the application of lagged temperature change relative to ice core records³⁶, benthic ocean temperature reconstructions⁴⁰, and ice sheet-proximal ocean temperature and salinity from global climate models^{32, 41}. Global climate models allow for spatial variability in ocean forcing for different regions of the ice sheet, which is advantageous. However, the TraCE-21ka simulation, which we use here, has been

suggested to show bias with respect to the oceanic response to prescribed deglacial meltwater^{42, 43}. In particular, because TraCE-21ka does not resolve the ice shelf cavity and relevant processes during its formation, we also performed modified ocean forcing experiments with ocean temperature anomalies imposed in the Ross Sea region during the Holocene, starting from 7 ka and finishing at the time of proposed ice sheet readvance (1.6 ka) by Neuhaus et al. (2021)¹².”

We also discuss the possible impact of higher variability in the ocean forcing in Lines 223 to 227 (Discussion):

“Our ISM simulations place an upper limit of ocean temperature anomalies in the cavity of 0.5°C averaged over 100 year time periods, though ocean forcing with higher variability may extend this temperature threshold higher. This means that even temporary intrusions of mCDW that increase the average temperature are sufficient for Holocene grounding line retreat beyond the present day. Such intrusions are more likely to occur under more stratified conditions with a shorter duration of winter sea ice and weaker katabatic winds (Fig 7c).”

Ultimately the authors employ two different approaches for the forcing

1. Ice core-based climate reconstructions + GCM ocean output.
2. The same but with synthetic ocean temperature anomalies.

Yes, and we now clarify our reasoning for doing so (see comment above).

Figure 2. Suggest adding the PD observed grounding line so the reader gets a bearing as to where the simulated grl. ends up in respect to the observed one. Additionally in panel (I) there seem to be either two 0ka grounding lines or it shows an almost fully collapsed WAIS.

The observed PD grounding line is shown in Fig 1 with the 0 ka configuration shown as the reference for WAIS retreat / advance in Fig 2. The model shows good agreement with the observed Siple Coast grounding line, but the position is generally advanced by 10s of km for the Mercer and Kamb Ice Streams. The simulation that the Reviewer refers to (panel I of previous draft) is unrealistic. The model is run with particularly high ocean thermal forcing as well as high mantle viscosity, resulting in significant ice sheet retreat in the WAIS subglacial basin, though partial re-grounding occurs as the ocean forcing cools. Given the Reviewer’s suggestions about both Fig 2 and Fig 3, we have removed the +0.6°C ocean forcing runs from these figures, and instead show the results in Fig S6.

I suggest changing the topography colormaps in the figures (e.g. to a grayscale) and plot the grounding line evolution in distinguishable colors.

We have changed the grounding line colors and added a key. We also eliminated a row, so the individual panels appear larger. While we have kept the topography colormap as is based on our preference, we do have a version of the figure in grayscale that we could use if this is the preferred choice of the Reviewer and/or editors:

Grayscale version of Fig 2:

The authors run their simulations on 20 km resolution, the lateral extend of the ice streams populating the Ross sector of the WAIS thus varies between ca. 1-2 grid boxes. This goes back to my previous comment on providing the modelled output at present day compared to observations. In this case, specifically the modelled surface velocity and thickness in the Ross Sector compared

to MeASURES and BedMACHINE (e.g. providing RMSE of the model ensemble and 2D vel.-difference plot for the best fitting ensemble member).

See above comments re: sensitivity experiments run with 10km resolution, and see Fig S3 for present-day simulation.

L88 how is an ice stream location defined in the proxy records and how do you compare it to the modelled location (midpoint of a central streamline?). Please elaborate.

We have clarified our ice stream location definition in the Fig 3 caption.

L103: as shown in Kingslake et al. 2018.

Kingslake et al. (2018) does not investigate the effect of anomalous mid-Holocene warming, late Holocene cooling.

Figure 3: What is the motivation to include the red line (simulations with +0.6C temperature anomaly)? This specific simulation suggests a collapsed WAIS at present day and thus should be considered unrealistic. I suggest removing it from the plot which would improve readability (lots of lines, colors, linestyles already). Also, where are the horizontal dashed lines? I don't see any. What is the meaning of the gray line (position of WIS and BIS). Is that the inferred paleo grounding zone? And if so, at which time? What are the colored vertical lines indicating? In general, I find this figure difficult to entangle. How do you define the grounding line position over time? Do you take a transient or constant central flowline and pinpoint the grounding line position along this flowline?

We have moved the simulations with +0.6°C ocean temperature anomalies to Fig S6. The intention is to demonstrate that we have explored the upper range of plausibility, though as Reviewer 3 notes, other basal melt parameterizations that are less sensitive to ocean temperature changes may have a higher threshold for ice shelf collapse. The horizontal lines correspond to the WIS and BIS site locations, with the vertical coloring indicating the timing of retreat and advance. We clarified our definition of the grounding line position in the caption.

Reference list: references are all small caps except first letter. Maybe an issue with the export from the respective bibliography-software.

Yes, this is from the bibliography style in use, which will be addressed in the production if accepted.

L111: ... has contributed to variable sea ...

Fixed.

L114: If I understand correctly the PISM simulations are used as an input for the GIA model. However, in the PISM output the Lingle-Clark bedrock deformation model is used so some kind of bedrock response to changing ice load is already included. This way you force GIA with data which already contains a bedrock response (albeit a less realistic one). I find this difficult to disentangle. But maybe I misunderstand something here.

Ice thickness from the ice sheet model is the only input into the global GIA models. With this in mind, we use the PISM simulations run with the Lingle-Clark Earth deformation model because these simulations produce realistic ice cover changes. Our previous work explored the effects of

global mean sea level forcing and Earth deformation on deglacial grounding line retreat in the Ross Sea (see Lowry et al., 2020 Fig 1 and Fig 7); as would be expected, neglecting these processes produces unrealistic ice sheet deglaciation that would not be useful to use as input into the GIA model. We also couple the ice sheet and GIA models offline using the iterative coupling approach of Gomez et al. (2018) for the simulations shown in Fig 6. Ideally all simulations would be fully coupled, but this is infeasible for the entire ensemble.

Figure 4: missing label for colorbar. I do not understand how bed topg. changes between 2-7 ka BP (rather 7-2ka BP) are illustrated. Is this a mean topg-change during this period? Or the bedrock response at the end of the period (2ka BP)? Also, your notation is a little confusing. I assume you always implicitly mean BP but then I'd write 7 – 2 ka BP instead of 2 – 7 ka.

A label for the color bar has been added and notation revised.

Supplementary figures: suggest checking the aspect ratio for all 2D plots. AIS looks a bit “squished”. Also please add complete label, now only unit is given (m). All figures add PD observed grounding line for reference.

Done.

L127: which ice cover changes. From this line it seems you focus on one ISM realization. Which one? This should get a dedicated figure (see comment above).

Figure reference added.

L146 I don't find any mention of what uplift would be significant for sub-ice shelf ocean circulation in Tinto et al. This brings me back to Figure 4. If you write bed topography changes for 0-2ka (2 – 0 ka BP), do you mean a temporal average? Or is this depicting the topo changes at PD? If so, shouldn't they be very close to zero? Topo changes of 100-300 meters would point to an inconsistent ice load history, wouldn't they?

This figure is showing the difference in bed topography between those time periods. The GIAWE model is especially sensitive to changes in ice loading; the ice thickness changes are shown in the Supplemental Information.

L151 effect on what? Response of what? Relative “cooling” or “warming”? Figure 3 shows that synthetic t-anomalies are all positive. Or do you refer to the standard forcing?

This has been rephrased.

L152 isn't it the other way around? Ice sheet thins and retreats due to increasing basal melt rates? Or do you refer to increases in melt rates due to deeper grounding lines i.e. lower melting point?

This has been rephrased.

L154 specify! Not everyone knows what the current melt rates in the Amundsen sea sector are. Also peak melt rates? Can be over 100 m/a underneath Thwaites or PIG, alternatively bulk rates (10-20 m/a).

We have specified that we are referring to bulk melt rates of 10-20 m/a.

Figure 5: for me it's rather difficult to identify which line color corresponds to what time. I

suggest using a more variable cmap and including a legend. The overlapping bedrock and lower ice shelf boundary make this plot difficult to read. Again, poor resolution, suggest using vector graphics or raster with ≥ 300 dpi.

We have adjusted the line colors and made the figure 300dpi.

L161 yes, increased snowfall can increase discharge, but this discharge does not offset the SMB increase completely (about 30-60% according to Winkelmann et al., 2012). Also, this refers to SMB changes integrated over the entire continent. Regional increase in SMB e.g. at the grounding line wouldn't increase the dynamic ice flux necessarily because they might even reduce surface gradients and thus lower driving forces.

This sentence has been removed.

L163 which, in our simulation/can/ lead(s) to an instantaneous ... in the transition between the retreat and advance phase.

Changed.

L164 Surface mass balance would require a 30-fold increase ... : increase with respect to what?

We have clarified we are referring to the present day.

L165 how does that relate to the findings in Neuhaus et al., 2021 who suggest that regional climate change (modest ocean cooling) was the culprit behind grounding line re-advance?

Our model results support the Neuhaus et al. (2021) hypothesis that modest ocean cooling is the primary cause of grounding line re-advance. We have added:

Line 198 to 201 (Results: Relative Contributions):

“Overall, these simulations demonstrate that the ice sheet response to relative cooling is robust and consistent with the timescales of change inferred from the age constraints of radiocarbon input and decay model, regardless of the bed topography changes over the past 2.0 ka. This result supports the Neuhaus et al., (2021)⁹ hypothesis that modest oceanic cooling is the driver of late Holocene ice sheet readvance in this sector of the WAIS.”

Paragraph 174-188: the ice sheet response to the different bedrock scenarios discussed in the text are difficult to spot in Fig. 6 due to the choice of cmaps. I would suggest picking a more distinguishable colormap with perceivable color/contrast changes for the grounding line positions at different times. For the bedrock anomaly cmap you could consider picking white for the transition across zero (e.g. simple RdBu cmap).

We have adjusted the grounding line colors and added a legend.

Case c) in Fig. 6 suggests substantial uplift underneath the present day ross ice shelf leading to ice rises/ice domes on islands. Wouldn't that be a strong indicator that the combination of ice load history and mantle viscosity is unrealistic and should be excluded? This should at least be mentioned (e.g. in line 179).

We have now clarified that this simulation is unrealistic. The GIAWE model is relevant from the standpoint that it uses an Earth structure comparable to that estimated for the Amundsen Sector of

West Antarctica. But these simulations demonstrate that this Earth structure is too weak for the Siple Coast and/or that 3D variations in Earth structure need to be considered. Weak Earth structure may be more relevant for the Western Ross Sea, where mantle viscosity is estimated to be relatively lower (Whitehouse et al., 2019), and the GIAWE shows a reasonable fit to the Scott Coast relative sea level reconstruction from Hall et al., (1999).

Lines 196 to 198 (Results: Relative Contributions):

“The high uplift in the GIAWE simulation grounds portions of the Ross Ice Shelf, highlighting the particular sensitivity to this weak Earth structure and suggesting such Earth structure is unrealistic for the Siple Coast of WAIS as well as the importance of considering the laterally varying (3D) Earth structure in Antarctica for robust comparison between model results and observations.”

Lines 256 to 261 (Discussion):

“While no single set of parameter values in these ice sheet and 1D GIA models can fully capture these ranges, the ensembles are useful for highlighting that mantle viscosity affects bed topography, which in turn influences ice sheet sensitivity to changes in oceanic conditions (Figs 2,3). In particular, the GIAWE simulation shows reasonable migration of the Siple Coast grounding line, whereas the GIAWE does not. This is relevant for the Siple Coast because it has relatively higher mantle viscosity and lithosphere flexural rigidity by West Antarctic standards, meaning that the grounding line here is more sensitive to increases in ocean thermal forcing than other areas of weaker mantle.”

L195 picking up the point from above, doesn't the strong bed uplift on the Ross shelf leading to extensive ice rises/domes at present day disqualify a GIAWE structure?

See above comment on GIAWE.

L209 deuterium isotopes

Corrected.

L238-240 A perspective on how (un)realistic the GIAWE results are in terms of ice history and bed rebound is missing (see comments above).

See above comment on GIAWE.

L255 please provide parameters chosen for Eigencalving and thickness calving here. Even better, provide table of key parameters/forcings changed in the ensemble (i.e. flow enhancement factors, basal substrate, and sliding parameters, calving approximations, glacioisostatic, adjustment, and surface climate and ocean thermal forcing) e.g. in the supplements.

Done. See Table S1.

L262 please specify how you construct the surface climate forcing? WAIS divide record is a point measurement, how do you translate this to a spatial forcing covering the entire ice sheet domain?

We apply a percentage anomaly relative to Van Wessem et al. (2014), which is now specified.

L268-270 please provide a comparison (supplements) of simulated pd melt rates and estimates

(e.g. Adumusilli et al. 2020.).

We have added the melt rates to Fig S3, which are consistent with estimates from Adumusilli et al. (2020).

Reviewer #3 (Remarks to the Author):

Lowry et al., present a proxy-informed series of ice sheet model simulations to investigate if Holocene grounding line (GL) retreat in the Siple Coast region was driven by climatic (ocean, atmospheric forcing) or non-climatic processes (glacioisostatic adjustment). This is very timely paper, with significant implications for ice sheet stability. It is also the first paper to try and attribute forcing to Holocene GL retreat/overshoot. I can't comment on the specifics of the modelling, but the approach seems well conceived and thorough. Generally, the paper is very well written with appropriate/informative figures.

My only substantive comment relates to the timing of change(s), and how they are presented in the MS. I think the authors can do a much better job in highlighting the uncertainty associated with the 'timing' of retreat and readvance (derived from published literature). Currently there's a very large window when retreat is thought to have occurred (quoted as 7.5 to 2.0 ka in the MS, although I think this should be 7.2 – 1.7 ka based on the Venturelli and Neuhaus papers), with readvance more tightly constrained to after ~1.5 ka (note that in Figure 7, the pink interval/readvance phase is more like ~2.3 ka to present and this should be updated or better justified). Furthermore, all these ages, especially those in Neuhaus have very large uncertainty attached to them. On the upper end, GL retreat at Whillans grounding zone and Mercer occurred at ~7.2 and 6.3 ka, respectively (Venturelli et al., 2020; 2023). Conversely, retreat on Bindschadler and Kamb was as late as 1.8 ka and 1.7 ka (Neuhaus et al., 2021), with readvance as recently as 0.8 ka. This results in a very large target to work with! In this respect, I wonder if the MS would be stronger if you explicitly set this up as sensitivity experiment rather than needing the model to replicate retreat/advance at a particular time? The large spread of GL retreat ages also creates problems relating to the forcing and how these model simulations fit with available proxy data/current thinking regarding ocean/atmosphere forcing i.e., it's very difficult to attribute a particular glacial response to a driver when the timing is so uncertain and variable across the Siple Coast. For e.g., Lowry et al suggest that a switch from a warm cavity to cold cavity can be used to explain retreat and re-advance, but proxy data imply that this transition occurred after 3.6 ka. Can this transition really explain GL retreat of Bindschadler and Kamb after 1.8 ka? Interestingly, Hall et al. (2006). PNAS., infer a warmer than present interval in the western Ross Sea ~2.3 to 1.1 14C kyr BP (Hall et al., 2006), which is consistent with the timing of retreat for Bindschadler and Kamb but not necessarily the 'cold cavity' scenario presented in the MS and summarised in Figure 7.

Thus, in summary, I think the revised MS needs to be much more precise/transparent regarding the timing of GL retreat – GL changes occurring at 7.2 ka are probably not driven by the same processes as changes at 1.7 ka. You could include this in the main body of the MS or as an extra section in the supplementary information. Given the large spread of GL retreat ages, I would also like to see more consideration about the factors driving the apparent spatial variability. Venturelli et al. (2020) speculated that differences in ocean forcing in the eastern and western Ross Sea might explain the discrepancy between their age (7.2 ka) and the earlier GL retreat age (9.7 ka) in Kingslake et al. (2018). Can you say anything more about this? Rather than try and say that warm/cold cavity transition explains all GL variability, I think it would be more instructive to set out which ice streams/ice sheet thinning histories do not fit this model?

We thank the Reviewer for their positive comments on our approach and presentation of results, as well as the helpful citations and clarifications on the timing of changes of the West Antarctic ice streams. These have significantly improved our presentation of results as well as our discussion in which we contextualise the modelling with the various proxies.

More specifically, we have added more detail to differences in timing of retreat / advance of WIS and MIS compared to KIS and BIS. In terms of the ice sheet model, our experiments support different sensitivities of these ice streams to changes in ocean forcing, as demonstrated by the more extensive retreat/advance of WIS compared to BIS in Fig 3 and 6 that occurs over a shorter time frame. In additional experiments, now shown in the Supplemental Information, we also demonstrate that a later cutoff of anomalous ocean temperature (e.g. 1 ka BP) produces improved agreement for the later timing of BIS and KIS indicated in Neuhaus et al. (2021).

We have included the citation of Hall et al. (2023) QSR, which updates and extends the work of Hall et al. (2006). In particular, they note warmer-than-present oceanic conditions in the Western Ross Sea from 7.1 to 0.5 ka BP, with peak warm periods occurring at 5.2 and 2.3-1.8 ka BP; the former corresponds to retreat phase of WIS, and the latter corresponds with the retreat phase of KIS and BIS. These data are incorporated in the revised Figure 7. We also note that Mezgec et al. (2017) discuss variability in polynya efficiency during the Late Holocene. Our proposed mechanism of WAIS advance from increased overturning / reduced mCDW intrusion is consistent with these various proxy records.

Lastly, as the Reviewer notes, the ice streams may have experienced differences in ocean forcing. We expanded our discussion of this point with regard to the results of the Tinto et al. (2019), which show different water masses interacting with the southern versus northern ice streams in their ocean model experiments. Differences in timing in the transition from mCDW to low and high salinity shelf water, and/or differences in water mass properties, may explain why advance in KIS and BIS occurred later. Additional cavity-resolving regional ocean model experiments could be used to test this hypothesis, but this is beyond the scope of the present study.

Here are our specific text revisions addressing these comments:

Lines 19 to 21 (Introduction):

“The timing of WAIS readvance in this sector has been inferred from radiocarbon input and decay modelling to have occurred within the past 1.7 ka for the WIS, and even more recently for the Kamb and Bindshadler ice streams (KIS and BIS, respectively)⁹.”

Lines 114 to 117 (Results: Ice sheet response to ocean forcing):

“Notably, the radiocarbon modelling indicates differences in behaviour between the various ice streams, with later retreat and advance of the more northern KIS and BIS relative to the southern WIS. If we apply the anomalous forcing to 1 ka BP, the timing of our modelled readvance of BIS and KIS is more consistent with the inferred ages of Neuhaus et al. (2021)⁹ (Fig S6).”

Lines 191 to 195 (Results: Relative Contributions):

“For the WIS and KIS, the ISM with simple viscoelastic Earth deformation, GIAWE, and GIASE show grounding line advance to near-modern position within 400 years, with advance of BIS delayed by another 200 years. This demonstrates the lower sensitivity of the BIS to changes in ocean forcing than the WIS, but can only partly account for the difference in timing indicated by

the radiocarbon input and decay model (Fig 3c). We discuss other possibilities for differences in ice stream behaviour in the following section.”

Lines 239 to 250 (Discussion):

“For the Siple Coast grounding line, the ice sheet simulations shown here demonstrate that a regime shift from a warm to cold ocean cavity is capable of driving hundreds of kilometers of advance. In terms of geologic evidence, subglacial precipitates also suggest that millennial-scale oscillations in Southern Ocean temperatures drive ice sheet velocity fluctuations through addition of subglacial meltwater from the ice sheet interior during warm periods and subglacial freezing during cold periods⁶⁴. Furthermore, centennial-to-millennial scale shifts in the El Niño Southern Oscillation and Southern Annular Mode are observed to have dramatically influenced circum-Antarctic winds and sea ice heterogeneously the Late Holocene^{65, 66}. Such shifts would influence heat flux in the cavity, and therefore raises the possibility of grounding line oscillations at these timescales within the Holocene. While the Neoglaciation transition represents a state change in Southern Ocean conditions on millennial timescales, evidence of centennial-scale Late Holocene variability in polynya efficiency is also evident¹⁹, with low polynya efficiency during a particularly warm phase from 2.3 to 1.8 ka BP indicated by the presence of elephant seals in the western Ross Sea⁴⁵, also coinciding with retreat of BIS and KIS9. WIS advances following this Late Holocene warm event, but BIS and KIS advance within the last 0.8 ka BP following strong regional cooling that led to collapse of the seal population (Fig 7a).”

Lines 262 to 274 (Discussion):

“Another influence of bed topography is on the sub-ice shelf ocean circulation, with the possibility that uplift of the bed contributed to a transition from a warm to cold ocean cavity. Tinto et al. (2019)¹⁸ show that intrusion of warm mCDW is limited under the ice shelf front in the modern bed configuration, isolating WAIS from oceanic heat. But in a glacial maximum configuration, mCDW interacts with the ice sheet at the continental shelf edge. At some point since the last deglaciation, basal melt rates proximal to WIS become influenced by cool HSSW produced by polynyas on one side of the tectonic boundary, whereas basal melt rates proximal to BIS and KIS become influenced by low-salinity ice shelf water on the other. The ISM indicates that the southern ice streams of WIS and MIS are more sensitive to changes in ocean forcing than the northern ice streams of KIS and BIS, but another factor contributing to the lag in northern ice stream advance could be these differences in shelf water mass composition, as well as the timing of when these two sides of the cavity become isolated from mCDW. Our simulations do not rule out that bed topographic changes indirectly contributed to WAIS retreat and advance by altering the delivery of mCDW to the WAIS ice streams. Cavity-resolving regional ocean model simulations could test the relative influences of Late Holocene climatic conditions and bed topography on the sub-ice shelf ocean circulation and water mass properties.”

Revised Fig 7:

Figure S6, which shows the influence of anomalous ocean forcing duration:

Minor comments:

Lines 29-30: I don't think its necessary to specify 'modified' (-CDW) unless you want to discuss on shelf properties/oceanography.

We understand the Reviewer's point here, since the paper is intended for a broad audience. However, it is more accurate to specify that the CDW is "modified" because we are focusing on the Ross Ice Shelf, a cold cavity in which the mCDW is cooler ($\sim 1^{\circ}\text{C}$) as compared to CDW.

Line 30: Walker et al is a great paper but an odd choice here. Jacobs et al., 1996. GRL; Jenkins et al., 2009. Nat Geo; Dutrieux et al., 2014. Science?

We thank the Reviewer for the suggestion and have added the Dutrieux et al. (2014) citation.

Lines 74-77: Bit of a cop out. Either the timing matters or it doesn't. Much of the discussion revolves around when retreat/readvance occurred and what drove these changes.

This sentence has been removed and replaced with a reference to Fig S6, which shows the impact of timing of onset and cutoff of the anomalous ocean thermal forcing (see above).

Line 96: Choice of +0.6 deg C simulation; if this leads to ice shelf collapse does this imply that your model is too sensitive to ocean forcing? I note in the methods that the 'high melt rate'

parameter/end member is derived from Thwaites/PIG but is this forcing realistic for the Ross Sea during the Holocene?

The basal melt parameterization was developed so that melt rates in cold cavities, like the Ross Ice Shelf, are much lower than warm cavities under present-day conditions (see Lines 300 to 303 in the Methods, and Gollledge et al. 2019 as a reference). Ice shelf collapse occurs only after >1000 years of +0.6°C ocean temperature anomalies, so the duration is an important component of this upper limit to ice shelf viability, which we now explain in the revised manuscript. We agree with the reviewer that less sensitive basal melt parameterizations may have a higher temperature threshold. Alternatively, increasing variability of the ocean temperature forcing could increase the threshold. We now discuss this in the Supplementary Information with respect to Fig S6.

Line 136: refs out of order. Ref 46 should be 47?

Fixed.

Lines 200-202: Re-word. Mid-Holocene peak temperatures don't explain retreat of KIS (1.8 ka) and BIS (1.7 ka), only WIS (4.7 ka). Also include age-data from Mercer.

Done.

Fig. 7. As previously noted, background colours (blue, yellow, pink), which correspond to developing, warm and cold cavities are too simplistic to explain variable timing of GL retreat? One implication could be that BIS and WIS are behaving differently from WIS/Whillans grounding zone and Mercer? Would be more interesting to speculate about what drove this? Lines 320 (refs): A couple of recent papers not cited but are relevant are: Jones et al., 2023 (<https://agupubs.onlinelibrary.wiley.com/doi/full/10.1029/2020GL091454>) also suggests enhanced ocean-driven melt in the Early-to-Mid Holocene.

Melis et al. (<https://jm.copernicus.org/articles/40/15/2021/jm-40-15-2021.pdf>) indicate stronger bottom currents in late Holocene.

Venturelli et al. (2023)

(<https://agupubs.onlinelibrary.wiley.com/doi/epdf/10.1029/2022AV000846>) provide constraints on the timing of GL retreat Mercer Ice Stream.

We thank the Reviewer for these references, which we have included in the revised manuscript. We have added a discussion of differences between the various ice streams (see above response). We have also modified Fig 7 to the above suggestion of a later ice sheet advance and discuss the possibility of late Holocene variability in polynya efficiency noted in Mezgec et al. (2017) as a possible mechanism for the behaviour of BIS and KIS, along with differences in water mass sources (e.g. Tinto et al., 2019).

REVIEWER COMMENTS

Reviewer #1 (Remarks to the Author):

This is my second round of review of the manuscript entitled, "Ocean cavity regime shift reversed West Antarctic grounding line retreat in the late Holocene" by Daniel P. Lowry and colleagues and in general I found the manuscript to be in an improved state from the original submission. The topic of grounding line retreat and re-advance during the Holocene is timely, and the findings of the manuscript work toward filling a gap in our knowledge in the mechanisms that drive the reversal of grounding line retreat. Here I detail a handful of suggestions of edits that I believe will improve the manuscript in its final form:

1. In the second sentence of the abstract, the sentence should read "Yet recent geologic and modeled evidence suggests" to better align with the constraints that the authors are tying their records to. This suggestion relates to my earlier review in which I point out that the conflation of geologic evidence with modeled evidence muddies the waters of constraints being used.
2. On this same theme, it should be further emphasized either in the title or abstract that this study is a modeling exercise. One simple place to do this would be in the abstract--in the sentence that begins "Our results", the authors could simply state, "The results of our model simulations indicate".
3. Line 18: The sentence that cites papers 4-6 should also include a citation to Balco et al. (2023; doi: 10.5194/tc-17-1787-2023), which provides evidence for a smaller-than-present configuration in the Amundsen Sea Sector. These results feed into some of the compilations cited, but should also be cited as they are "emerging" as mentioned earlier in the sentence.
4. Discussion of Ross Ice Shelf's positive mass balance and relevance would benefit from linking this work to the ISMIP 6 work, notably Seroussi et al. (2023; doi: 10.5194/tc-17-5197-2023) and (2020; 10.5194/tc-14-3033-2020). The first place the exclusion of these references struck me was on lines 27-29, but the authors should review throughout how these references could be worked in.
5. Line 70: Suggest using "updated" instead of "corrected" to remove the negative connotation/implication that previous work was wrong.
6. Line 74: The final sentence of this paragraph that reads, "Surface climate and ocean fields are updated every 100 years" should be clarified. At present it is not clear what this refers to.
7. Line 90-91: The sentence here (and the findings throughout) indicate that *both* mantle viscosity and ocean thermal forcing exert strong control over grounding line behavior in the Holocene, yet the headline style findings reported in the title and abstract only discuss the role of ocean thermal forcing. This disagreement suggests that some nuance should be added to the title to more adequately reflect the findings of this study.
8. Line 100 should read grounding-line-proximal AND SUBGLACIAL as Venturelli et al 2020 is GL-proximal, while 2023 is fully subglacial. Also suggest revision of Line 103—the presentation of the uncertainty range here does not match the final paper or the presentation of other uncertainty ranges in the paper.
9. Paragraph from 111-121: The presentation of "more northern" to differentiate between sites is inherently confusing. Suggest instead adopting a "further inland/inboard", "nearer the grounding zone" description if trying to differentiate between differences in proximity to the grounding line, or describing which portion of the basin the ice streams flow into if that is the important piece here. As written, it is unclear of what importance the inclusion of direction is playing.

10: Line 146: I still do not understand the logic of aligning the model simulation in this work with a modeled outcome (Neuhaus et al., 2021) when geologic constraints that are tighter than that modeled outcome exist. The authors do a great job incorporating those geologic constraints elsewhere in the manuscript, but it is unclear why the focus here still seems to overlook those. I suggest adding a bit more description here to demonstrate that the timing is not being selected to align with the desired outcome of the manuscript.

11: Paragraph beginning on 152: When discussing sides of the Ross Embayment, it is more precise to refer to the Transantarctic Mountain side as the "Western Ross Sea" than "the other side." Suggest being specific here, because it is confusing as written.

12: Paragraph beginning on line 175: There is quite a lot of data on surface mass balance beyond what is being presented here in a Bodart et al (2023; 10.5194/tc-17-1497-2023). The discussion comparing the work herein to changes in surface mass balance would really benefit this work. I realize that throughout this manuscript a common theme of my comments have been to relate this work to more updated references, but given the timeliness of this topic I think it is of utmost importance to ensure that the references cited accurately reflect the state of knowledge on Holocene retreat and re-advance.

13. Line 213: correct grammar in statement "too gradual than"

14. Discussion of subglacial precipitates on lines 244-246 seems irrelevant and out of place given those archives cover a much longer timescale than the work presented herein. Suggest either removing reference to this work or adding further discussion of how this work is connected.

15. Line 253, BIS, KIS, and WIS are discuss, but MIS is excluded. MIS should be discussed here too.

16. Paragraph beginning on line 266: The impacts of the tectonic boundary on ocean-driven grounding line retreat was first discussed in Venturelli et al (2020) and then again in Neuhaus et al (2021). Some connection should be made between the inferences in those papers and this paragraph. Further, I would suggest looking into the updated work of Tankersley et al (2022; doi: 10.1029/2021GL097371) which builds on the Tinto et al (2019) work referenced here. Updated insight into sub-RIS structure could help to strengthen, clarify, and improve the discussion of ocean processes herein.

Figures:

Figure 1: It should be clarified in this figure where age constraints come from direct measurements (WGZ; MIS) and modeled results (BIS, KIS, WIS).

Figure 3, 4g/e, 5, 7: When presenting age on the x-axis, labeling years before present as negative is confusing. The presentation of older (left) to present (0/right) already implies that the paper is talking about the past. Suggest removing negatives to align with other papers about the Holocene.

Figure 3: To better align with adjustments made to the manuscript text in referring to recent publications excluded from the original submission (e.g., Venturelli et al., 2023), the timing of retreat of Mercer Ice Stream should be added to this figure

Reviewer #2 (Remarks to the Author):

The authors have addressed most of my comments and the manuscript is in good shape for publication. There is however one exception to this being that the authors don't really show the associated uncertainties and biases with respect to their model-setup. While I appreciate the addition

of figure S3 this doesn't really address my question. The authors state in their response:

More recent versions of the VUW-PISM model under present-day conditions are shown in Golledge et al. (2019) (PISM 0.7.3; See Fig 1 and Extended Data Fig 7), Seroussi et al., (2020) (PISM 0.7.3; see Fig 1 and 3), and Lowry et al. (2021) (PISM v1.1.2; see Fig S1 and S2).

This is all great but the authors use PISM2.0 (considerable changes since V0.7.3 and also different enough from 1.1.2 to merit a repeated assessment of model behavior under control conditions). Figure S3 merely shows the ice sheet response to climate conditions during 1950 – 2015 i.e. 65 years. This is not enough to assess whether the ice sheet is in equilibrium at the start of their simulation or to judge the trend and biases. If the authors amend this figure by a transient plot of sea level equivalent ice volume and mass change this would already suffice.

On another note, the authors state in the manuscript that the SLE in their spinup run is "within 4% of the total SLE ice volume of Antarctica". This doesn't square with what is shown in figure S3. Thus, I assume that the ice sheet has not equilibrated at this stage. Given that the authors assess the grounding zone evolution in a relatively narrow band of the Ross Sea sector during the Holocene they should show convincingly that for PD conditions their ice sheet setup is adequate to resolve such regional processes.

To reiterate my previous assessment I think showing the robustness of their model setup for PD/PI would greatly improve the manuscript and allow the reader to interpret the results in light of model uncertainties. This does not mean I doubt the quality of the model results presented here but merely reflects that all continental scale model exercises at this resolution suffer from biases. As the authors state, their RMSE sits square within the ISMIP6 range.

We thank the three reviewers for their time and constructive reviews of our revised manuscript, which have further improved our work. Our responses to each point are below in blue.

Reviewer #1 (Remarks to the Author):

This is my second round of review of the manuscript entitled, “Ocean cavity regime shift reversed West Antarctic grounding line retreat in the late Holocene” by Daniel P. Lowry and colleagues and in general I found the manuscript to be in an improved state from the original submission. The topic of grounding line retreat and re-advance during the Holocene is timely, and the findings of the manuscript work toward filling a gap in our knowledge in the mechanisms that drive the reversal of grounding line retreat. Here I detail a handful of suggestions of edits that I believe will improve the manuscript in its final form:

1. In the second sentence of the abstract, the sentence should read “Yet recent geologic and modeled evidence suggests” to better align with the constraints that the authors are tying their records to. This suggestion relates to my earlier review in which I point out that the conflation of geologic evidence with modeled evidence muddies the waters of constraints being used.

We have incorporated this change in the abstract as suggested.

2. On this same theme, it should be further emphasized either in the title or abstract that this study is a modeling exercise. One simple place to do this would be in the abstract--in the sentence that begins “Our results”, the authors could simply state, “The results of our model simulations indicate”.

We have changed “Our results” to “Our model results”.

3. Line 18: The sentence that cites papers 4-6 should also include a citation to Balco et al. (2023; doi 10.5194/tc-17-1787-2023), which provides evidence for a smaller-than-present configuration in the Amundsen Sea Sector. These results feed into some of the compilations cited, but should also be cited as they are “emerging” as mentioned earlier in the sentence.

We have added this citation.

4. Discussion of Ross Ice Shelf’s positive mass balance and relevance would benefit from linking this work to the ISMIP 6 work, notably Seroussi et al. (2023; doi: 10.5194/tc-17-5197-2023) and (2020; 10.5194/tc-14-3033-2020). The first place the exclusion of these references struck me was on lines 27-29, but the authors should review throughout how these references could be worked in.

We have revised as follows, citing Seroussi et al., (2023):

“Compared to other regions of WAIS, the Ross Embayment has been relatively stable during the observational era, with a positive mass balance^{14, 15}, though ice sheet model projections indicate this sector is vulnerable to future ocean warming¹⁶.”

5. Line 70: Suggest using “updated” instead of “corrected” to remove the negative connotation/implication that previous work was wrong.

PISM versions prior to PISM 1.1 had a flawed numerical implementation of the elastic component of the Earth deformation model, but this bug has since been fixed. We cite Albrecht et al. (2020), which explains this.

6. Line 74: The final sentence of this paragraph that reads, “Surface climate and ocean fields are updated every 100 years” should be clarified. At present it is not clear what this refers to.

We have revised as “These surface climate and ocean fields...” to clarify that we are referring to the surface climate and ocean fields we describe in the previous sentences.

7. Line 90-91: The sentence here (and the findings throughout) indicate that *both* mantle viscosity and ocean thermal forcing exert strong control over grounding line behavior in the Holocene, yet the headline style findings reported in the title and abstract only discuss the role of ocean thermal forcing. This disagreement suggests that some nuance should be added to the title to more adequately reflect the findings of this study.

We agree with the reviewer that this is an important point and have added the following to the abstract:

“Simulations with mantle viscosity in the higher range of estimates for West Antarctica show more rapid and extensive ice sheet retreat in response to ocean warming, but also show extensive centennial-scale readvance in response to abrupt ocean cooling. Because the Siple Coast region has higher mantle viscosity than other areas of West Antarctica, this implies that the grounding line here is sensitive to future changes in the Ross Sea polynya and sub-ice shelf ocean circulation.”

8. Line 100 should read grounding-line-proximal AND SUBGLACIAL as Venturelli et al 2020 is GL-proximal, while 2023 is fully subglacial. Also suggest revision of Line 103—the presentation of the uncertainty range here does not match the final paper or the presentation of other uncertainty ranges in the paper.

We have added “and subglacial”.

9: Paragraph from 111-121: The presentation of “more northern” to differentiate between sites is inherently confusing. Suggest instead adopting a “further inland/inboard”, “nearer the grounding zone” description if trying to differentiate between differences in proximity to the grounding line, or describing which portion of the basin the ice streams flow into if that is the important piece here. As written, it is unclear of what importance the inclusion of direction is playing.

The terms “northern” and “southern” have been removed.

10: Line 146: I still do not understand the logic of aligning the model simulation in this work with a modeled outcome (Neuhaus et al., 2021) when geologic constraints that are tighter than that modeled outcome exist. The authors do a great job incorporating those geologic constraints elsewhere in the manuscript, but it is unclear why the focus here still seems to overlook those. I suggest adding a bit more description here to demonstrate that the timing is not being selected to align with the desired outcome of the manuscript.

We have revised as follows:

“We focus on this particular simulation because it is the most consistent with the timing of WIS and MIS retreat reconstructed in Venturelli et al. (2020)⁸ and (2023)⁹ as well as the timing of WIS, KIS and BIS retreat and advance modeled in Neuhaus et al. (2021)¹⁰, while also maintaining a stable ice shelf.”

11: Paragraph beginning on 152: When discussing sides of the Ross Embayment, it is more precise to refer to the Transantarctic Mountain side as the “Western Ross Sea” than “the other side.” Suggest being specific here, because it is confusing as written.

We have changed “the other side of the embayment” to “the western Ross Sea side of the embayment”.

12: Paragraph beginning on line 175: There is quite a lot of data on surface mass balance beyond what is being presented here in a Bodart et al (2023; 10.5194/tc-17-1497-2023). The discussion comparing the work herein to changes in surface mass balance would really benefit this work. I realize that throughout this manuscript a common theme of my comments have been to relate this work to more updated references, but given the timeliness of this topic I think it is of utmost importance to ensure that the references cited accurately reflect the state of knowledge on Holocene retreat and re-advance.

We also now include this reference as follows:

“Furthermore, internal reflective horizons detected by radio-echo sounding have been used to infer that surface mass balance was 18% higher relative to modern over the WAIS Divide at 4.7 ka BP⁵⁸, coinciding with the radiocarbon-modeled retreat of WIS, KIS, and BIS¹⁰.”

Also, with regard to surface mass balance, we do show the WAIS Divide accumulation record in Fig S1 over a longer time frame (20 to 0 ka BP).

13. Line 213: correct grammar in statement “too gradual than”

We have changed to “more gradual than inferred...”

14. Discussion of subglacial precipitates on lines 244-246 seems irrelevant and out of place given those archives cover a much longer timescale than the work presented herein. Suggest either removing reference to this work or adding further discussion of how this work is connected.

We have added to this paragraph to clarify our point that, over a range of timescales, the geologic evidence points to ocean forcing as a key control on ice sheet changes, which is consistent with our ice sheet / GIA model results:

“For the Siple Coast grounding line, the ice sheet simulations shown here demonstrate that a regime shift from a warm to cold ocean cavity is capable of driving hundreds of kilometers of advance, consistent with geological evidence over a range of timescales. For example, subglacial precipitates also suggest that millennial-scale oscillations in Southern Ocean temperatures drive ice sheet velocity fluctuations through addition of subglacial meltwater from the ice sheet interior during warm periods and subglacial freezing during cold periods⁶⁷.”

Furthermore, centennial-to-millennial scale shifts in the El Niño Southern Oscillation and Southern Annular Mode are observed to have dramatically influenced circum-Antarctic winds and sea ice heterogeneously the Late Holocene, as inferred from marine sediment cores^{68, 69}. Such shifts would influence heat flux in the cavity, and therefore raises the possibility of grounding line oscillations at these timescales within the Holocene.”

15. Line 253, BIS, KIS, and WIS are discuss, but MIS is excluded. MIS should be discussed here too.

This is because Neuhaus et al. (2021) does not provide an estimate for MIS readvance. To clarify this point, we have added the following:

“The timing of readvance of MIS is unconstrained, though the ice sheet model shows a synchronous timing of advance as WIS.”

16. Paragraph beginning on line 266: The impacts of the tectonic boundary on ocean-driven grounding line retreat was first discussed in Venturelli et al (2020) and then again in Neuhaus et al (2021). Some connection should be made between the inferences in those papers and this paragraph. Further, I would suggest looking into the updated work of Tankersley et al (2022; doi: 10.1029/2021GL097371) which builds on the Tinto et al (2019) work referenced here. Updated insight into sub-RIS structure could help to strengthen, clarify, and improve the discussion of ocean processes herein.

We have added the citation as follows:

“The Siple Coast’s graben-bounding faults have been argued to accommodate the glacioisostatic rebound following the last deglaciation⁷².”

Figures:

Figure 1: It should be clarified in this figure where age constraints come from direct measurements (WGZ; MIS) and modeled results (BIS, KIS, WIS).

We have outlined the magenta circles at the WGZ and MIS sites in black and clarify in the caption that this indicates the age constraints at these sites are from direct measurements, whereas the un-outlined magenta circles indicate modelled age constraints.

Figure 3, 4g/e, 5, 7: When presenting age on the x-axis, labeling years before present as negative is confusing. The presentation of older (left) to present (0/right) already implies that the paper is talking about the past. Suggest removing negatives to align with other papers about the Holocene.

Done.

Figure 3: To better align with adjustments made to the manuscript text in referring to recent publications excluded from the original submission (e.g., Venturelli et al., 2023), the timing of retreat of Mercer Ice Stream should be added to this figure

For our preferred presentation of the results, we have kept the figure showing the two ice streams WIS and BIS. This limits plot busyness (please note the previous review of Reviewer

2) as well as maintains consistency with Fig 5. Given the proximity of the MIS to WIS, we note that grounding-line migration occurs synchronously along these two ice streams in our ice sheet model, as demonstrated by the various simulations shown in Figure 2, so additional information would not be gained from MIS inclusion in the plot. BIS shows different timing of retreat and readvance in both the radiocarbon model of Neuhaus et al. (2021) as well as the ice sheet model here and is a particular point of focus in our discussion (please note the previous review of Reviewer 3). With regard to MIS, we now refer the reader to Fig 2h in this sentence to highlight the timing of MIS retreat occurs after 5 ka BP for this particular simulation, later than the age control of Venturelli et al. (2023). We explain that a higher or earlier warm ocean forcing, or higher mantle viscosity, leads to earlier retreat of this site.

Reviewer #2 (Remarks to the Author):

The authors have addressed most of my comments and the manuscript is in good shape for publication. There is however one exception to this being that the authors don't really show the associated uncertainties and biases with respect to their model-setup. While I appreciate the addition of figure S3 this doesn't really address my question. The authors state in their response:

More recent versions of the VUW-PISM model under present-day conditions are shown in Golledge et al. (2019) (PISM 0.7.3; See Fig 1 and Extended Data Fig 7), Seroussi et al., (2020) (PISM 0.7.3; see Fig 1 and 3), and Lowry et al. (2021) (PISM v1.1.2; see Fig S1 and S2).

This is all great but the authors use PISM2.0 (considerable changes since V0.7.3 and also different enough from 1.1.2 to merit an repeated assessment of model behavior under control conditions). Figure S3 merely shows the ice sheet response to climate conditions during 1950 – 2015 i.e. 65 years. This is not enough to assess whether the ice sheet is in equilibrium at the start of their simulation or to judge the trend and biases. If the authors amend this figure by a transient plot of sea level equivalent ice volume and mass change this would already suffice.

On another note, the authors state in the manuscript that the SLE in their spinup run is “within 4% of the total SLE ice volume of Antarctica”. This doesn't square with what is shown in figure S3. Thus, I assume that the ice sheet has not equilibrated at this stage. Given that the authors assess the grounding zone evolution in a relatively narrow band of the Ross Sea sector during the Holocene they should show convincingly that for PD conditions their ice sheet setup is adequate to resolve such regional processes.

To reiterate my previous assessment I think showing the robustness of their model setup for PD/PI would greatly improve the manuscript and allow the reader to interpret the results in light of model uncertainties. This does not mean I doubt the quality of the model results presented here but merely reflects that all continental scale model exercises at this resolution suffer from biases. As the authors state, their RMSE sits square within the ISMIP6 range.

We fully agree with the Reviewer that model spin-up is important due to its impacts on model behaviour. Because the previous versions of the manuscript and our response referenced other studies, this may have led to some confusion about our approach. We have addressed this in this newly revised version of the manuscript, firstly, by including the following information about our spin-up procedure in the Methods section:

“We initiate our ISM experiments with a long spin-up procedure, the purpose of which is to allow the thermal structure of the ice sheets to evolve over a sufficiently long period that even deep layers are influenced by changing atmospheric conditions. We begin with an initial 20-yr smoothing in which only the Shallow Ice Approximation is used. Then, holding ice geometry fixed, we perform a 130-kyr thermal equilibration run using paleoclimate temperature forcing from the EPICA Dome C ice core (Jouzel et al., 2007), applied as an anomaly to the modern climatology (Van Wessem et al., 2014). We then perform a 130-kyr run in which we implement full model physics. Lastly, to ensure that our initial model state accurately reproduces glacial conditions at the start of the transient deglacial simulations, we precede each simulation with a 20 kyr period during which a constant “glacial maximum” climate field (i.e. 20 ka BP) is applied.”

As a point of clarification to the Reviewer, at no point in our study do we state our spin-up run is “within 4% of the total ice volume of Antarctica”, nor is this stated in the studies we cite in our methodology (i.e. Golledge et al., 2019; 2021). Bedmachine-Antarctica has an uncertainty range of +/- 0.9 m for SLE ice volume, and our spin-up simulation is within this range. Ice thickness error relative to Bedmachine-Antarctica from the end of our historical simulation at 2015 is shown in Fig S3b (see next page).

Lastly, as suggested by the reviewer, we have amended Figure S3 to include a transient plot of the sea level equivalent ice volume change (Fig S3f), which shows minimal model drift through the historical simulation (see next page). We also show the ice volume changes of all deglacial experiments in Figure S2a.

Fig S3. Present-day ice sheet model reference simulation. This reference simulation was run for the period 1950 to 2015 using the model parameters listed in Table S1 and climate forcing derived from NorESM1-M. Shown are a) modelled ice thickness at 2015 and (b) the differences to Morlighem et al. (2020); c) modelled ice surface velocity at 2015 and d) difference to Mouginit et al. (2017); e) Modelled basal melt. The black lines indicate the modelled ice sheet grounding and calving line positions, whereas the purple lines in the difference plots indicate the observed grounding and calving line positions. f) Transient ice volume change (sea level equivalent in m) relative to 1950 of the reference simulation.

Reviewer #3

The revised submission by Lowry et al. is improved, especially when discussing the timing of GL retreat based on published literature. The authors also explicitly discuss the factors which might explain differences in timing/forcings of GL retreat for each ice stream and incorporate relevant missing literature.

Occasionally details are still incorrect (Line 20: Should be 7.5 to 6.3 ka BP NOT 5.3 ka BP) or vague (line 22....within the past 1.7 ka for the WIS, and even more recently for the Kamb and Bindschadler ice streams (KIS and BIS, respectively). For the latter, I'd quote the modeled ages for KIS and BIS here, especially as this becomes important when talking about the factors (and timing) of readvance. Similarly, when talking about ocean proxies i.e., line 88, I'd again be more being specific. Hall et al. (2023) use the age-distribution of elephant seals to infer sea-ice variability and rely on other proxy data e.g., Xu et al. (2021). <https://agupubs.onlinelibrary.wiley.com/doi/10.1029/2021GL094545>, to infer intrusion of CDW. Additionally, while Hall et al. (2023) indicate that the largest reduction in sea-ice occurs at ~5.2 ka and between 2.3 and 1.8 ka, Xu et al. suggest more/warmer CDW 6.0-2.8 ka and 1.6-0.7 ka, with reduced CDW 2.8-1.8 ka. You note some of this detail on lines 242 and in Fig. 7 so I'd be specific from the start (rather than 'proxies support intrusion of mCDW between 7.2-0.5 ka BP').

The 5.3 ka BP accounts for the younger age-bound of Venturelli et al. (2023), i.e. considering the uncertainty of 1.0 ka.

We have revised the introduction paragraph to include the BIS and KIS readvance ages:

“even more recently for the Kamb ice stream (KIS) and Bindschadler ice stream (BIS), at 1 ka BP and 0.8 ka BP, respectively.”

We thank the Reviewer for recommending the Xu et al. (2021) paper. We have revised that sentence as follows by including that reference:

“As proxies from the Ross Sea region support enhanced intrusion of mCDW from 6 to 2.8 ka BP and 1.6 to 0.7 ka BP⁴⁷, the aim of these sensitivity experiments is to assess the grounding line response to plausible changes in Holocene oceanic forcing.”

More generally – and like my comment relating to the original submission - I think the warm/cold cavity model as presented in Fig. 7 is too simplistic (and unnecessary). You basically end up saying that centennial-scale polynya variability in the Late Holocene (2.3-1.8 ka) resulted in a warm cavity which drove retreat of BIS and KIS.....but the MS text and Fig. 7 discusses the presence of warm and cold cavities at 7.5-3.6 ka and 3.6-0 ka respectively. Since you state there is uncertainty in the timing of cold cavity initiation (Fig. 7 caption) why not steer clear of these arbitrary regimes and simply have oscillating cold/warm cavities if that's what's supported by proxy data. Fig. 7 would then have a yellow strip 2.3-1.8 ka and back to pink after 1.8 ka? Following this suggestion means that there's no need to have the overlapping regions or link the cartoons (bottom panel Fig.7) to the top panel with lines (I think the colored boxes are sufficient).

The purpose of Figure 7 is to show how our model experiments relate to the overall paleoclimate evolution of the Ross Sea region, as reconstructed by proxies, and serves as the

basis for our discussion section. More specifically, the ocean forcing we apply to the ice sheet model is intended to represent specific processes (e.g. enhanced mCDW intrusion, polynya overturning), albeit in simplistic fashion, in order to test how the ice sheet responds. It is our view, therefore, that this is important to include because it serves as a visual aid to readers as we discuss the various proxy records and relate them to ice sheet behaviour.

We agree with the Reviewer that we do need to account for the complexity of the processes driving ice sheet evolution, as well as their associated timescales. First, we do not have constraints for how far inland the ice sheet grounding line retreated, nor the precise timing of when ice sheet advance began. The marine records offshore Adélie Land indicate the initiation of cold cavity-like processes (e.g. enhanced production of supercooled shelf waters and an increase in sea ice production) at the start of the Neoglaciation, at 4.5 ka BP (Ashley et al., 2021; Johnson et al., 2021). However, as the Reviewer notes, there is variability in terms of polynya efficiency and mCDW intrusion indicated in other Western Ross Sea records (e.g. Xu et al., 2021). Furthermore, differences in ice stream behaviour may point to regional variation in cavity transition timing. Our intention in including the overlapping yellow and pink color in Figure 7 is to add more nuance to the discussion of when the cavity transition occurred. We have revised the caption to more explicitly explain this point:

“The overlapping yellow and pink represents the uncertainty in the precise timing of cold cavity initiation and with acknowledgment that regional differences may exist. The timing is based on the long-term increase in Ross Sea polynya efficiency from 3.6 ka BP²¹, though centennial-scale oscillations in polynya efficiency and mCDW intrusion do occur through this interval^{47, 59}, until the time of ice stream advance indicated in pink from 1.6 ka BP.”

As suggested, we have also removed the lines linking the ice sheet schematics to the top panel of proxy records.

Figure 3: I note the discussion about the +0.6degC simulation in the rebuttal (in response to #Reviewer 2).....but how/why has the +0.5deg C ocean forcing curve simply ‘replaced’ the +0.6deg C forcing curve in Figure 3? It appears to be the same line but now labelled +0.5? Does that mean the original figure was incorrect or have I missed something?

The line for the +0.5°C simulation is accurate in all submitted versions of the manuscript. You can see the comparison to the +0.6°C line in Fig S6. The +0.6°C simulation has an earlier and more extensive grounding line retreat of WIS and BIS.

REVIEWER COMMENTS

Reviewer #1 (Remarks to the Author):

The authors have responded adequately to all of my comments, and I have no further edits, questions, or clarifications to make. Thank you to the author team and editor for considering my input, I hope it was helpful. Congratulations on a truly wonderful piece of work!

Reviewer #2 (Remarks to the Author):

This is now the 3rd round of reviews. I appreciate the clarification of the paleo spinup which is thorough and well suited for this kind of study. I have no further comments on that.

That being said, I still have trouble understanding what's going on in Figure S3. The authors provide the PISM parameters in their reference run in Table S1. If I'd plug in these parameters for a historical simulation I would get ice thickness changes of up to 1000 m in West Antarctica. You can have a look e.g. at the supplements of Reese et al 2023 who use PISM and go through great lengths to optimize their setup for present day conditions. They still get thickness changes in the year 2015 (end of historical) in West Antarctica of several hundred meters. How did you start the historical reference simulation? Bootstrapping from Bedmachine/Bedmap2? Or starting from your free evolution 130 kyr run? Do you use some kind of iterative friction optimization? Your table suggests you use the standard topg2phi parameterization. Figure S3. Also the pattern of thickness changes is not something you'd expect from this kind of "flow law" in PISM (pseudo plastic + till friction angle).

I assume, that the reason for the small changes in thickness is that you run the historical control simulation only for ~60 years. Even with 1-3 m/a thickness changes (rates observed today for e.g. Thwaites see Smith et al. 2020, Science Figure 3) you'd only get 60 - 180 meter surface lowering. The historical reference run you provide only shows the initial response to the forcing. You'd have to run this simulation for several hundred years or more to get the equilibrium response under historical or pi conditions.

Given that this is already the 3rd round of reviews, I'll leave these comments here as merely a suggestion to further clarify the present day equilibrium response of PISM to the reference parameter setting.

This is now the 3rd round of reviews. I appreciate the clarification of the paleo spinup which is thorough and well suited for this kind of study. I have no further comments on that.

That being said, I still have trouble understanding what's going on in Figure S3. The authors provide the PISM parameters in their reference run in Table S1. If I'd plug in these parameters for a historical simulation I would get ice thickness changes of up to 1000 m in West Antarctica. You can have a look e.g. at the supplements of Reese et al 2023 who use PISM and go through great lengths to optimize their setup for present day conditions. They still get thickness changes in the year 2015 (end of historical) in West Antarctica of several hundred meters. How did you start the historical reference simulation? Bootstrapping from Bedmachine/Bedmap2? Or starting from your free evolution 130 kyr run? Do you use some kind of iterative friction optimization? Your table suggests you use the standard topg2phi parameterization. Figure S3. Also the pattern of thickness changes is not something you'd expect from this kind of "flow law" in PISM (pseudo plastic + till friction angle).

I assume, that the reason for the small changes in thickness is that you run the historical control simulation only for ~60 years. Even with 1-3 m/a thickness changes (rates observed today for e.g. Thwaites see Smith et al. 2020, Science Figure 3) you'd only get 60 - 180 meter surface lowering. The historical reference run you provide only shows the initial response to the forcing. You'd have to run this simulation for several hundred years or more to get the equilibrium response under historical or pi conditions.

Given that this is already the 3rd round of reviews, I'll leave these comments here as merely a suggestion to further clarify the present day equilibrium response of PISM to the reference parameter setting.

We thank the reviewer for these suggestions, which we have taken on board in our revised Supplemental Information for this manuscript.

We appreciate the reference to Reese et al. (2023), which performed a comprehensive analysis of present-day grounding line dynamics using a wide range of PISM parameter combinations. However, it should be noted that the goals of these two studies differ, which explains differences both in terms of PISM setup and our approach to model validation.

In our case, we are focused on deglacial ice sheet changes of a particular region, i.e. the Ross Embayment, and model parameters are optimised for that purpose. Our assessment of model parameter combinations places emphasis on the regional LGM ice thickness and extent, timing and magnitude of deglacial ice thinning, and timing of grounding line retreat, as we explain in the Supplemental Information and demonstrate with Figure S2. The two studies also have notable differences in terms of basal melt parameterisation and model resolution (Reese et al. is run at 8 km horizontal resolution vs. 20 km in our case), both of which would strongly influence best-fit parameter combinations. We explain the latter with respect to the 10 km resolution simulation we perform:

“Because increasing model resolution increases ice sheet and ice shelf velocity and produces thinner ice shelves, we ran this experiment using lower enhancement factors for the Shallow Ice and Shallow Shelf Approximations”

In terms of the till friction angle, we do use the standard topg_to_phi parameterization in PISM. We have revised the ice thickness difference plot in Fig S3 because there previously

was a slight grid offset between the ice sheet model and the Bedmachine ice thickness, which likely led to the Reviewer's concerns regarding the ice thickness change. This has been corrected (see figure on next page).

To further clarify our historical reference simulation, we have revised the Supplemental Information as follows, confirming that we are bootstrapping from Bedmachine using the output from the model spin-up:

“We also run a present-day reference simulation, in which we map ice temperature fields, internal velocities and bed conditions from our model spin-up to the present-day ice sheet configuration (Morlighem et al. 2020), and run forward in time using historical climate forcing from NorESM1-M (Bentson et al., 2013). This produces reasonable fit to observations of grounding line position, ice thickness (Fig S3a,b) and ice surface velocity (Fig S3c,d). Bias with respect to ice surface velocity does occur at the Siple Coast of Antarctica, with lower-than-observed surface velocity of West Antarctic ice streams and regions of higher-than-observed ice shelf velocity. Modern basal melt rates are within the ranges estimated by Adusumilli et al. (2020), with modelled melt rates as high as 18 m yr^{-1} in the Amundsen Sea sector ice shelves, but generally $< 1 \text{ m yr}^{-1}$ for the Ross Ice shelf, which overlies a cold ocean cavity (Fig S3e). We extend this run for 500 years to demonstrate the ice sheet response to present-day climate over a longer duration. Sea level equivalent ice volume shows a small decrease (Fig S3f), as expected given modern Antarctic mass loss, but grounding line position remains relatively unchanged at the Siple Coast (gold line in Fig S3b,d).”

We have also revised Figure S3 to show the transient ice volume change and grounding line position under present-day climate for this simulation, now extended to 500 years (see next page):

Revised caption:

“Fig S3. Present-day ice sheet model reference simulation. This reference simulation was run for the period 1950 to 2450 using the model parameters listed in Table S1 and climate forcing derived from NorESM1-M. Following 2015, the climate forcing is held constant using a mean present-day climate. Shown are a) modelled ice thickness at 2015 and (b) the differences to Morlighem et al. (2020); c) modelled ice surface velocity at 2015 and d) difference to Mouginot et al. (2017); e) Modelled basal melt. The black lines indicate the modelled ice sheet grounding and calving line positions at 2015, the dark gold line indicates the modelled grounding position at 2450, and the purple lines in the difference plots indicate the observed grounding and calving line positions. f) Transient ice volume change (sea level equivalent in m) relative to 1950 of the reference simulation.”